# Chemical warfare between leafcutter ant symbionts and a co-evolved pathogen

Daniel Heine[1], Neil A. Holmes[2], Sarah F. Worsley[2], Ana Carolina A. Santos[3,4,5], Tabitha M. Innocent[6], Kirstin Scherlach[3], Elaine H. Patrick[2], Douglas W. Yu[2], J. Colin Murrell[7], Paulo C. Vieria[5], Jacobus J. Boomsma[6], Christian Hertweck[3,4], Matthew I. Hutchings[2] & Barrie Wilkinson[1]

*Acromyrmex* leafcutter ants form a mutually beneficial symbiosis with the fungus *Leucoagaricus gongylophorus* and with *Pseudonocardia* bacteria. Both are vertically transmitted and actively maintained by the ants. The fungus garden is manured with freshly cut leaves and provides the sole food for the ant larvae, while *Pseudonocardia* cultures are reared on the ant-cuticle and make antifungal metabolites to help protect the cultivar against disease. If left unchecked, specialized parasitic *Escovopsis* fungi can overrun the fungus garden and lead to colony collapse. We report that *Escovopsis* upregulates the production of two specialized metabolites when it infects the cultivar. These compounds inhibit *Pseudonocardia* and one, shearinine D, also reduces worker behavioral defenses and is ultimately lethal when it accumulates in ant tissues. Our results are consistent with an active evolutionary arms race between *Pseudonocardia* and *Escovopsis*, which modifies both bacterial and behavioral defenses such that colony collapse is unavoidable once *Escovopsis* infections escalate.

[1] Department of Molecular Microbiology, John Innes Centre, Norwich Research Park, Norwich, Norfolk NR4 7UH, UK. [2] School of Biological Sciences, University of East Anglia, Norwich Research Park, Norwich, Norfolk NR4 7TJ, UK. [3] Leibniz Institute for Natural Product Research and Infection Biology, HKI, Beutenbergstraße 11a, Jena 07745, Germany. [4] Friedrich Schiller University, Jena, Germany. [5] Departmento de Química, Universidade Federal de São Carlos, UFSCar, Via Washington Luiz KM 235, CP 676, São Carlos, SP, Brazil. [6] Department of Biology, Centre for Social Evolution, University of Copenhagen, Universitetsparken 15, Copenhagen 2100, Denmark. [7] School of Environmental Sciences, University of East Anglia, Norwich Research Park, Norwich, Norfolk NR4 7TJ, UK. These authors contributed equally: Daniel Heine, Neil A. Holmes. Correspondence and requests for materials should be addressed to M.I.H. (email: m.hutchings@uea.ac.uk) or to B.W. (email: barrie.wilkinson@jic.ac.uk)

Ant colonies have nested levels of immune defense encompassing a lower level (individual ants) and a higher collective level that is usually referred to as social immunity[1]. These social immune defenses are so efficient that specialized epidemic ant diseases are generally unknown[2,3]. This suggests that essentially no pathogens have been encouraged over evolutionary time to specialize on ants as hosts, except when they could evolve mechanisms to make infected ants leave the protective social immunity of their colonies. Well-known examples are the *Ophiocordyceps* and *Pandora* fungi that turn infected ants into zombies and make them die in remote places that are optimal for pathogen spore dispersal[4–7]. Consistent with this impressive general immune-efficiency, there are no specialized ant diseases known in the attine fungus-farming ants, but their cultivar gardens are plagued by a single specialized fungal parasite *Escovopsis*[8] which arose simultaneously with ant agriculture 55–60 million years ago[9].

*Acromyrmex* leafcutter ants form a tripartite superorganismal mutualism with the fungal cultivar *Leucoagaricus gongylophorus* and a defensive actinobacterial symbiont *Pseudonocardia* (Fig. 1)[10]. The cultivar is a functionally polyploid clone[11] that the ants provision with freshly cut leaves and carefully groom to remove spores and hyphal growth of *Escovopsis* and generalist fungal pathogens[12]. In return for food and housing, the cultivar has evolved specialized hyphal structures, called gongylidia[13] which are rich in lipids and sugars. The gongylidia serve two functions: they are the sole food source for the ant larvae but are also ingested by the ant workers to transmit fungal decomposition enzymes to the ant fecal fluid which is deposited on the fresh leaf pulp[14]. Very early in their evolutionary history, the attine ants started to rear biofilms of antibiotic-producing actinobacteria on their cuticles, which has generally been interpreted as a

specialized defense against *Escovopsis*[15]. However, these biofilms do not all have *Pseudonocardia*[16] and were not universally maintained over evolutionary time as some attine genera lost the cuticular actinobacteria[17,18]. The most striking contrast of this kind exists between the two genera of leafcutter ants, where *Atta* species use phenylacetic acid from their metapleural gland secretions to control *Escovopsis*[17], while *Acromyrmex* species maintain the cuticular biofilms that are known to reduce the prevalence of *Escovopsis*[12]. The same cuticular biofilms may also have sanitation benefits for the ant brood (Fig. 1)[16,19–23].

The leafcutter ant symbiosis has been particularly well studied in the Panamanian species *Acromyrmex echinatior* where the cuticular biofilm is dominated by either *Pseudonocardia octospinosus* or *Pseudonocardia echinatior*, two distinct species of actinobacteria with ca. 50/50 population-wide prevalence[24,25]. These *Pseudonocardia* species are predicted to make different variants of the broad-spectrum polyene antifungal nystatin[26,27], similar to nystatin-like compounds that have been isolated from *Pseudonocardia* mutualists cultured by *Acromyrmex octospinosus* ants collected in Trinidad and a basal attine species *Apterostigma dentigerum* in Costa Rica[20,28,29]. The *Apterostigma* mutualist strains also make a cyclic depsipeptide called dentigerumycin which has antifungal activity against *Escovopsis*[30].

It has been estimated that about 50% of *A. echinatior* nests in Panama are colonized by *Escovopsis*[31] and previous studies have provided largely anecdotal evidence that worker ants die or abandon their gardens when the cultivar is overrun by *Escovopsis*[8]. In *A. echinatior*, such events reflect a failure of the *Pseudonocardia* biofilm defenses. They occur frequently when colonies are freshly excavated and kept in lab nests where the ants cannot dump their waste away from the garden. The resulting colony collapse suggests that *Escovopsis* hyphae produce

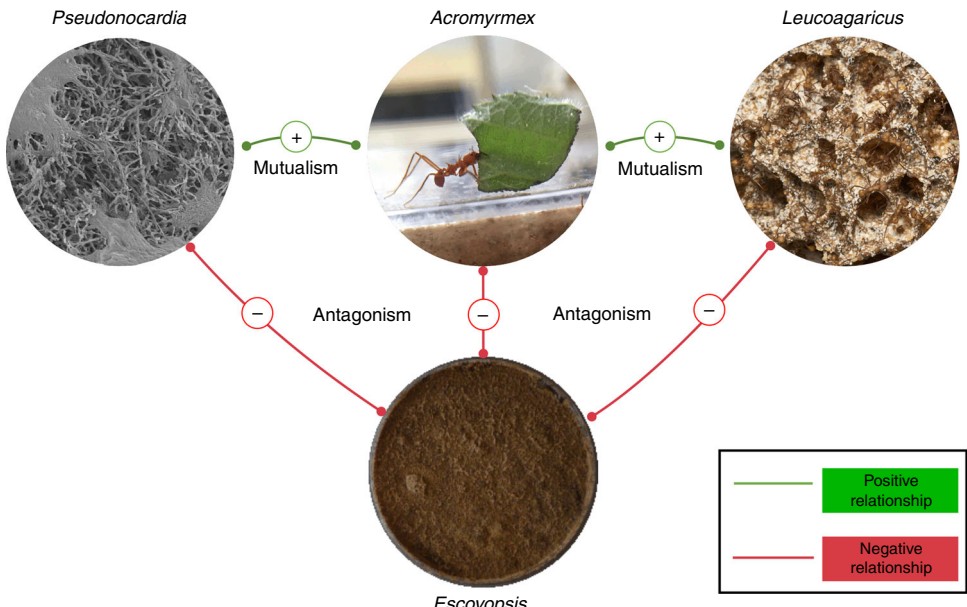

**Fig. 1** Summary of the complex interactions between symbionts of *Acromyrmex echinatior* leafcutter ants. The ants feed their vertically transmitted bacterial mutualist strain of *Pseudonocardia* through tiny subcuticular glands and in return the bacteria provide antifungal compounds to kill the parasite *Escovopsis*. They feed cut leaf fragments to their clonal fungal mutualist *Leucoagaricus gongylophorus* and this fungus provides the sole food source for their larvae. The parasite *Escovopsis* also feeds off *Leucoagaricus* and if left unchecked can overrun and kill the fungal cultivar and the entire ant colony. To prevent this, worker ants groom their fungal cultivar to remove *Escovopsis* spores and sterilize them with antifungal compounds made by *Pseudonocardia*. In this work, we show that *Escovopsis* fights back against the defensive mutualists by producing the virulence factors melinacidin IV (**1**) and shearinine D (**2**). Both compounds kill *Pseudonocardia* and **2** also adversely affects worker ant behavior and is ultimately lethal to the ants. The SEM of *Pseudonocardia* sp. spores on the surface of *A. echinatior* ant was taken by Dr. Kim Findlay (JIC); the images of the *Acromyrmex* ant and fungus garden were taken by Professor Matt Hutchings (UEA); the image of the *Escovopsis* plate was taken by Dr. Neil Holmes (UEA)

metabolites that affect worker ant behavior and which, at high enough concentrations, are lethal to the ants. However, in unstressed colonies the cuticular *Pseudonocardia* defenses are highly effective, such that it is usually impossible to isolate *Escovopsis* from long-term laboratory colonies maintained under optimal conditions. *P. octospinosus* and *P. echinatior* also produce a multitude of antibacterial compounds, most likely to defend their position in the cuticular biofilms against secondary infections[26,27,32], a metabolic activity that may also explain their general sanitation effects towards ant brood[19]. These interactions suggest that the *Acromyrmex* leafcutter ant symbiosis is the result of complex arms races between specialized pathogens and a collective of mutualists[33]. However, the chemical details of these interactions remain severely understudied.

Here, we identify two secondary metabolites that are upregulated during *Escovopsis weberi* infection of the fungal cultivar and show that they target the defensive *Pseudonocardia* mutualists in this system. We report that melinacidin IV (**1**) and shearinine D (**2**) are overproduced during infection and both kill the two *Pseudonocardia* species associated with Panamanian *A. echinatior* colonies. **2** is a terpene-indole alkaloid closely related to the penitrems which have been linked to the behavioral changes in carpenter ants infected with the *Ophiocordyceps* 'zombie ant' fungus[34]. We show that ingestion of purified **2** by garden worker ants adversely affects their behavior and is ultimately lethal to the ants. Furthermore, we show that worker ants from fungus gardens artificially or naturally infected with *E. weberi* contain levels of **2** significantly above those of controls. We sequenced and compared the genomes of five *Escovopsis* strains isolated from *Atta* and *Acromyrmex* leafcutter ant nests with a previously published genome sequence from an *Atta*-associated strain and found they all encode shearinine-like biosynthetic gene clusters (BGCs). We hypothesize that shearinines are produced to impede the ability of leafcutter worker ants to efficiently groom and weed their fungus gardens.

## Results

**Upregulation of *Escovopsis* virulence factors.** The pathogenesis of *Escovopsis* on the fungal cultivar *L. gongylophorus* was studied by co-culturing with *E. weberi* strain G (Supplementary Table 1) on potato glucose agar (PGA) plates and comparing the secondary metabolome to that of axenic cultures of *E. weberi*. Chemical profiling of extracts taken from the resulting plates, using ultra performance liquid chromatography (UPLC) coupled with high-resolution mass spectrometry (HRMS), revealed the presence of two major, and a range of minor, metabolites produced during pathogenesis. The major metabolites had signals at $m/z$ 729.0937 ($[M+H]^+$) for compound **1** and 600.3340 ($[M+H]^+$) for compound **2** (Fig. 2). Using liquid chromatography (ultraviolet)-mass spectrometry (LC(UV)MS) analysis calibrated using isolated standards (see Supplementary Notes 1 and 2), we confirmed that **1** and **2** were elevated significantly (3.4-fold for **1** and 8.9-fold for **2**) when compared to axenic cultures of *E. weberi* (Fig. 2). Similar results were obtained for the pathogenesis of *E. weberi* strain A when carried out independently, with upregulation of compounds **1** (2.2-fold) and **2** (2.9-fold) during pathogenesis, and we further confirmed that **1** and **2** were absent in extracts of the healthy food fungus (Supplementary Fig. 1).

**Identification of the *Escovopsis* virulence factors.** For structure elucidation we isolated **1** and **2** from large-scale axenic cultures of *E. weberi* strain A grown on PGA plates. Ethyl acetate extraction, silica-based column chromatography and semi-preparative high-performance liquid chromatography (HPLC) yielded pure samples of **1** (2.0 mg) and **2** (1.4 mg). HRMS analysis of **1** showed an intense M+2 signal, indicating a number of incorporated sulfur atoms. Only half the expected number of signals could be observed in the $^1$H and $^{13}$C nuclear magnetic resonance (NMR) spectrum, indicating a symmetrical compound. Combining these data, a molecular formula of $C_{30}H_{28}N_6O_8S_4$ was derived. Analysis of 2D NMR data (heteronuclear single-quantum correlation

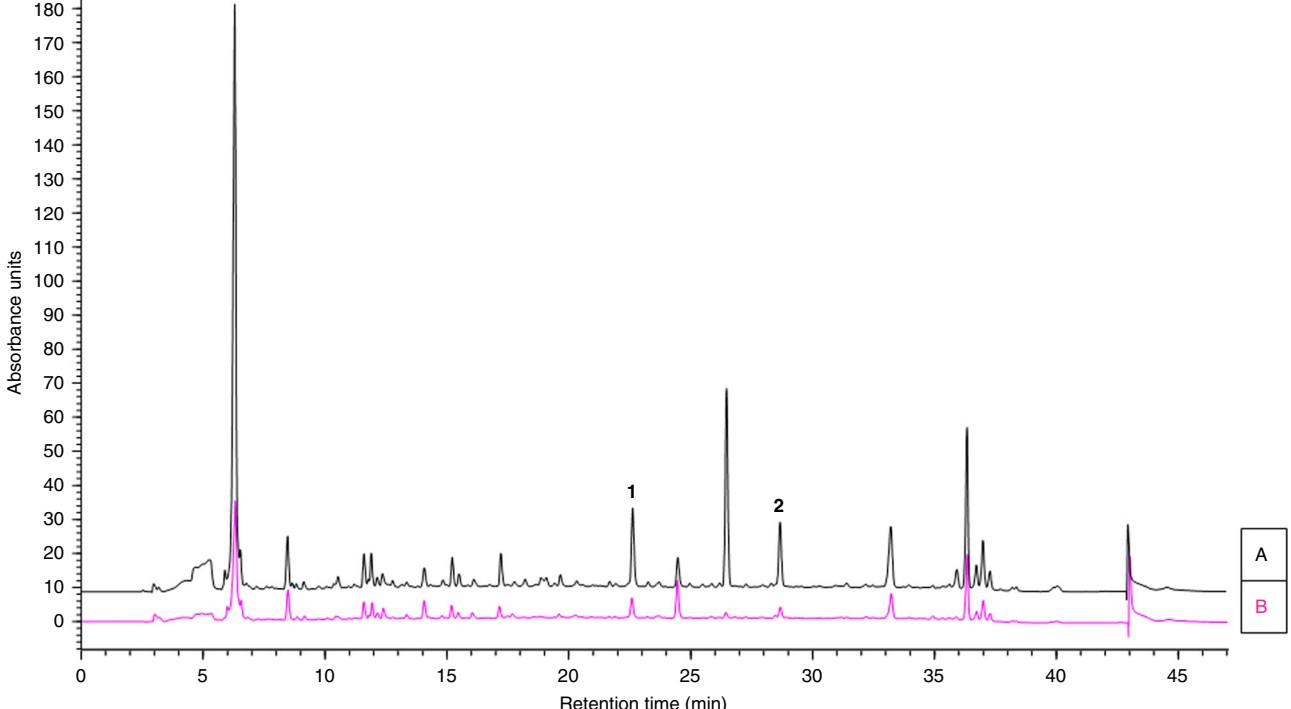

**Fig. 2** *Escovopsis* pathogenesis involves upregulation of small molecule virulence factors. HPLC profiles ($\lambda = 254$ nm) of ethyl acetate extracts of **a** *L. gongylophorus* infected with *E. weberi* strain G; **b** *E. weberi* strain G axenic culture. Melinacidin IV, **1**; shearinine D, **2**

**Fig. 3** Chemical structures of the major metabolites produced by *Escovopsis* strains

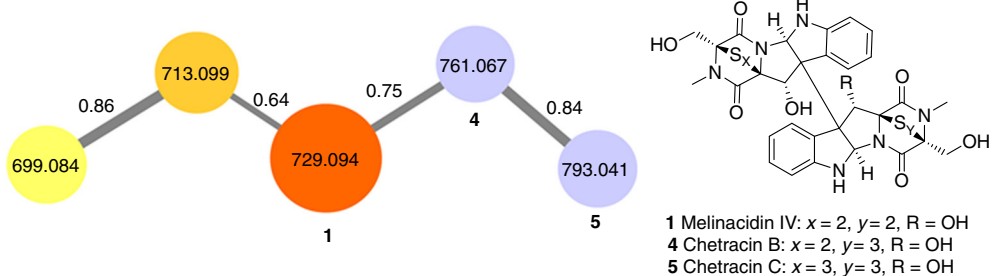

**Fig. 4** Molecular network and chemical structures of ETPs being produced by *Escovopsis* strains A–F

(HSQC) and heteronuclear multiple-bond correlation (HMBC)) pointed to a 1,2-substituted phenyl-ring as part of an indole moiety, while the second part of the molecule was clearly comprised of a diketopiperazine. Comparison of all analytical data, including optical rotation with literature values[35], finally confirmed **1** as the epipolythiodiketopiperazine (ETP) antibiotic melinacidin IV, originally isolated from the fungus *Acrostalagmus cinnabarinus* var. melinacidinus[36] (Fig. 3; Supplementary Note 1).

The [1]H NMR spectrum of **2** showed 4 aromatic protons, 14 aliphatic protons and 24 protons that could be assigned to 8 methyl groups. Further, we determined 16 quaternary carbon atoms, including 1 carbonyl group by analysis of the [13]C NMR spectrum. Accordingly, we could assign a molecular formula of $C_{37}H_{45}NO_6$, featuring 16 double bond equivalents. Structure elucidation based on 2D NMR data (HSQC and HMBC) revealed an indole moiety as part of a highly condensed carbon backbone and a Michael acceptor system in proximity to an acetal unit. These features along with the comparison of the chemical shift of all NMR signals with literature data, nuclear Overhauser effect spectroscopy (NOESY) analysis, and the measurement of the optical rotation unambiguously confirmed the identity of **2** as shearinine D, a terpene-indole alkaloid previously isolated from the endophytic fungus *Penicillium janthinellum* (Fig. 3; Supplementary Fig. 2; Supplementary Note 2)[37]. Predicted BGCs for terpene-indole alkaloid and ETP metabolites are present in the published *E. weberi* genome of strain G[38] (Supplementary Figs 3 and 4; Supplementary Tables 2-7), and shearinines D, F and J have been identified using imaging mass spectrometry of the

*Escovopsis* strain TZ49 isolated from *Trachymyrmex zeteki* nests from the same Panamanian field site[39]. No function was assigned to these compounds in the previous study and the species of *Escovopsis* was not identified[39].

**Metabolomics and molecular networking**. Having identified two major classes of secondary metabolites upregulated during *E. weberi* infection of the fungal cultivar, we clarified their distribution among *Escovopsis* lineages from different attine hosts. We tested five additional *Escovopsis* strains isolated from the nests of the leafcutter ants *Atta colombica* (strain C), *Acromyrmex echinatior* (strains B, E and F), and the higher, non-leafcutter attine *Trachymyrmex cornetzi* (strain D) (Supplementary Table 1). We cultivated five replicates of each strain, plus strain A as a control, and performed UPLC-MS/MS-based profiling. Data were uploaded to the Global Natural Product Social Molecular Networking (GNPS) web platform and used to perform a molecular networking analysis approach[40]. Molecular networking captures the similarity of analytes by the comparison of their MS/MS spectra and this allowed us to identify a number of metabolite families being produced by the *Escovopsis* strains (Supplementary Fig. 5). Molecular networking revealed that additional congeners of **1** are present in all six strains (Figs. 4 and 5). We observed compound **3**, putatively annotated as melinacidin III (*m/z* 713.0969 [M+H]+, molecular formula of $C_{30}H_{28}N_6O_7S_4$; Supplementary Note 3) and a further derivative of melinacidin III, missing one methyl group (*m/z* 699.0866

| ETPs Retention time (RT; min) | Positive mode $m/z$, $[M+H]^+$ | Sum formula M | ID | Escovopsis strains | | | | | |
|---|---|---|---|---|---|---|---|---|---|
| | | | | A | B | C | D | E | F |
| 4.25–4.41 | 699.0886 | $C_{29}H_{26}N_6O_7S_4$ | Unknown | 5 | 5.7 | 5.6 | 0 | 5.6 | 5.7 |
| 4.32–4.58 | 713.0972 | $C_{30}H_{28}N_6O_7S_4$ | Melinacidin III (3)* | 5.3 | 6.2 | 5.6 | 5.4 | 6 | 6 |
| 4.37–4.74 | 729.0934 | $C_{30}H_{28}N_6O_8S_4$ | Melinacidin IV (1)* | 5.9 | 6.5 | 6.7 | 5.7 | 6.6 | 6.6 |
| 4.38–4.59 | 761.0672 | $C_{30}H_{28}N_6O_8S_5$ | Chetracin B (4)** | 5.8 | 5.3 | 5 | 5.4 | 5.6 | 5.4 |
| 4.41–4.62 | 793.0316 | $C_{30}H_{28}N_6O_8S_6$ | Chetracin C (5)** | 5.5 | 0 | 0 | 2.9 | 0 | 0 |

| Terpene-indole alkaloids RT (min) | Negative mode $[M-H]^-$ $m/z$ | Sum formula M | ID | Escovopsis strains | | | | | |
|---|---|---|---|---|---|---|---|---|---|
| | | | | A | B | C | D | E | F |
| 6.13–6.20 | 486.3032 | $C_{32}H_{41}NO_3$ | Unknown | 2.1 | 6.6 | 0 | 0 | 6.4 | 6.3 |
| 4.66–4.94 | 498.31 | $C_{33}H_{41}NO_3$ | Unknown | 0 | 7.1 | 0 | 0 | 7 | 6.9 |
| 4.70–4.94 | 500.3249 | $C_{33}H_{43}NO_3$ | Unknown | 0 | 6.9 | 0 | 0 | 6.8 | 6.7 |
| 4.50–4.75 | 514.3035 | $C_{33}H_{41}NO_4$ | Unknown | 0 | 7.6 | 0 | 0 | 7.5 | 7.4 |
| 5.40–5.55 | 518.3285 | $C_{33}H_{45}NO_4$ | Sespendole* | 0 | 4.1 | 0 | 0 | 1.9 | 0 |
| 4.87–5.28 | 532.3097 | $C_{33}H_{43}NO_5$ | Unknown | 6.6 | 6.9 | 6.5 | 0 | 6.9 | 6.7 |
| 5.69–6.04 | 566.3348 | $C_{37}H_{45}NO_4$ | Penitrem D* | 6 | 6 | 4 | 0 | 3.8 | 3.9 |
| 4.83–5.10 | 580.3084 | $C_{37}H_{43}NO_5$ | 22,23-dehydro-shearinine A (7)** | 0 | 5.9 | 3.8 | 0 | 6.2 | 4 |
| 5.06–5.61 | 582.324 | $C_{37}H_{45}NO_5$ | Shearinine F* | 6.8 | 7.2 | 6.7 | 0 | 7.2 | 7.2 |
| 5.98–6.20 | 582.3245 | $C_{37}H_{45}NO_5$ | Shearinine A (6)** | 6.6 | 7.2 | 6.8 | 0 | 7.1 | 7 |
| 4.58–5.14 | 598.317 | $C_{37}H_{45}NO_6$ | Shearinine D (2)** | 7.3 | 7.7 | 7.6 | 0 | 7.7 | 7.6 |
| 4.81–4.92 | 600.3233 | $C_{37}H_{47}NO_6$ | Shearinine J* | 0 | 6.7 | 6.6 | 0 | 6.7 | 3.3 |

| 0 | 1 | 2 | 3 | 4 | 5 | 5.5 | 6 | 7 | 7.5 | 8 |
|---|---|---|---|---|---|---|---|---|---|---|

**Fig. 5** List of selected signals assigned to ETPs and shearinine-like terpene-indole alkaloid metabolites. Includes a heat map of their average ion count in logarithmic representation. *Metabolite putatively annotated; **metabolite identified and authenticated against isolated sample

$[M+H]^+$, molecular formula of $C_{29}H_{26}N_6O_7S_4$). Two related signals could be annotated as the ETPs chetracin B (**4**; $m/z$ 761.0670 $[M+H]^+$, molecular formula of $C_{30}H_{28}N_6O_8S_5$; Supplementary Note 4) and chetracin C (**5**; $m/z$ 793.0639 $[M+H]^+$, molecular formula of $C_{30}H_{28}N_6O_8S_6$; Supplementary Note 5) which was verified by NMR[41]. ETPs **4** and **5** feature polysulfide bridges of their diketopiperazine moiety of various lengths and are present in lower amounts in the extracts[42] (Fig. 3).

We also identified a complex network of shearinine-like terpene-indole alkaloid congeners (Fig. 5; Supplementary Fig. 6). Production levels were sufficiently high to allow isolation of several compounds from a scaled-up culture of strain C which resulted in **2** (3.5 mg), along with **6** (shearinine A; 1.0 mg) and **7** (22,23-dehydro-shearinine A; 1.8 mg), in addition to the polyketide metabolite emodin (**8**; 0.85 mg) (Fig. 3). Compound **6** showed an HRMS signal of $m/z = 584.3379$ corresponding to a

molecular formula of $C_{37}H_{45}NO_5$. Comparison of $^1H$ NMR, $MS^2$ data and the retention time of **6** with an authentic reference confirmed its identity as shearinine A (Supplementary Note 6). Compound **7** featured an HRMS signal of $m/z = 582.3209$ $[M+H]^+$ (corresponding to a molecular formula of $C_{37}H_{43}NO_5$) and could be identified as 22,23-dehydro-shearinine A by the comparison of $^1H$ and $^{13}C$ NMR data (Supplementary Note 7). Additionally, we observed the presence of likely pathway intermediates such as putative shearinine J and penitrem D (Fig. 5). The discovery of **8** (Supplementary Note 8) as a secondary metabolite produced by an *Escovopsis* sp. is highly intriguing. Anthraquinones have a long history as a feeding deterrent for ants[43–45] and **8** in particular has proven to be a broad-spectrum insecticide comprising activity against three different mosquito species[46], the white fly *Bemisia tabaci*[47] and caterpillar larvae[48]. Analysis of the sequenced genomes (see below) showed that all the *Escovopsis* strains contain the biosynthetic genes required for production of **8** (Supplementary Fig. 7; Supplementary Tables 8-10).

To determine the distribution and relative levels of the annotated metabolites among the different *Escovopsis* strains, we used the software Profiling Solutions (Shimadzu) for data analysis, peak picking, data alignment and filtering of metabolic profiling data. A heat map was generated showing the abundance of each ion in the MS profile (Fig. 5). The metabolomics data clearly showed that ETPs are produced by all six *Escovopsis* strains A–F. In contrast, shearinines are only produced by the five strains isolated from leafcutter ant nests (*Atta* and *Acromyrmex* species) but not by *Escovopsis* isolated from the higher, non-leafcutter attine ant *T. cornetzi* (strain D) (Fig. 5). We validated our metabolomics data through genome sequencing of strains A–F, which verified that a shearinine-like BGC is present in strains A–C and E–F (Supplementary Fig. 3; Supplementary Tables 2-4); however the BGC was absent from strain D.

**Genome sequencing and analysis of *Escovopsis* strains**. To gain deeper insight into the secondary metabolism of the fungal pathogen, we sequenced the genomes of the six *Escovopsis* strains A–F used in this study. The genome characteristics of these *Escovopsis* strains are shown in Supplementary Table 1. As a reference, we used the published genome of an *E. weberi* strain (strain G) isolated from *Atta cephalotes* collected in Gamboa, Panama; the same field site used to isolate strains A–F and for the

collection of ant colonies[38]. According to Meirelles et al.[49], there are nine clades of *Escovopsis*, with five classified species. Phylogenetic analysis (Supplementary Fig. 8, Supplementary Table 11) using the *tef1* gene and internal transcriber spacer (ITS) DNA sequences shows that strains A and C most closely resemble *E. weberi* species from clade I, and strains B, E and F most closely resemble *E. weberi* species from clade II. Strain D is distantly related to the *E. weberi* strains and most closely resembles *Escovopsis aspergilloides* and aligns to clade VII[49]. We analyzed all the sequenced *Escovopsis* genomes using fungiSMASH, an online resource for the rapid identification of natural product BGCs from fungi[50]. We detected 20–23 putative BGCs for each strain and these mainly comprise terpene, type 1 polyketide synthase and non-ribosomal peptide synthetase clusters. Based on the reported BGC for **2** in *P. janthinellum*[51], we identified homologous BGCs in strains A–C and E–G (Supplementary Fig. 3; Supplementary Tables 2-4). This is consistent with the fact that these strains all make shearinine-like terpene-indole alkaloids, but strain D does not. BGCs for the biosynthesis of ETPs like **1** are present in all seven genomes (Supplementary Fig. 4; Supplementary Tables 5-7) and resemble the BGC responsible for production of the ETP chaetocin from *Chaetomium virescens*[52].

**Virulence factors 1 and 2 inhibit *Pseudonocardia* mutualists**. Melinacidins are known to have potent activity against Gram-positive bacteria and we reasoned it would benefit the parasite to inhibit the *Pseudonocardia* mutualists[36]. As in some more basal attine branches[16], the *Acromyrmex* have a tight association with these bacteria as the ants have specialized crypts with small glands for housing and feeding the *Pseudonocardia* mutualist[15]. To test our hypothesis, we determined the minimum inhibitory concentration (MIC) of compounds **1** and **2** against representative strains from the two lineages of *Pseudonocardia* associated with *A. echinatior* colonies collected in Gamboa, Panama. Strain Ae707 belongs to the Ps1 lineage (*P. octospinosus*) and strain Ae706 belongs to the Ps2 lineage (*P. echinatior*)[27]. Liquid-based colorimetric assays were performed in microtiter plates to determine the MIC. Ciprofloxacin was used as a positive control and all the assays were done in triplicate. The results show that compounds **1** and **2** are active against both tested *Pseudonocardia* mutualist strains. Compound **1** has an MIC of 10 µg ml$^{-1}$ against Ae706 and 0.5 µg ml$^{-1}$ against Ae707, while compound **2** has an MIC of 5 µg ml$^{-1}$ against both Ae706 and Ae707 (Fig. 6).

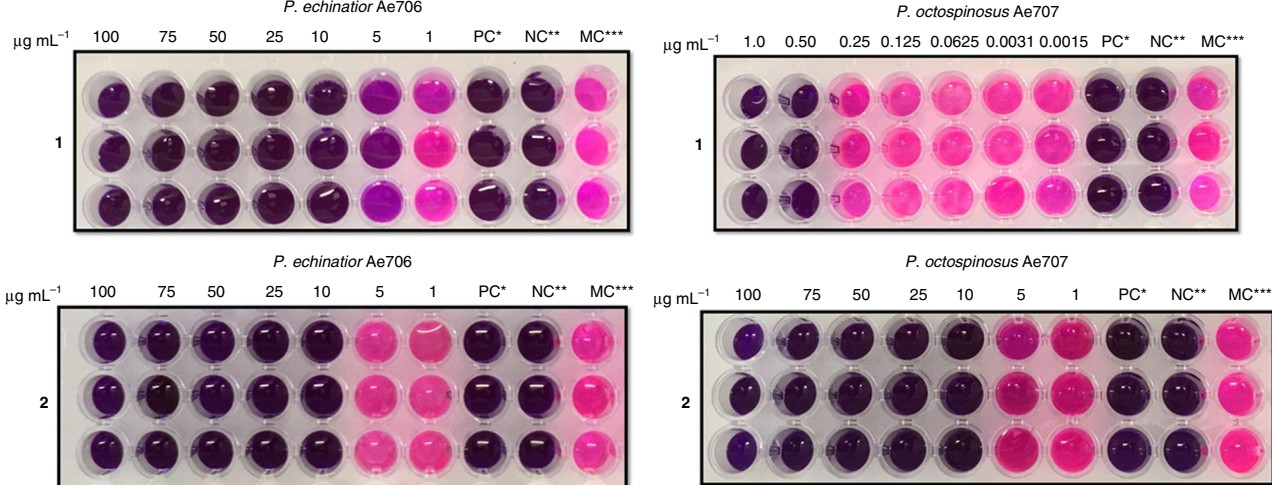

**Fig. 6** Growth of *Pseudonocardia* mutualists is inhibited by *Escovopsis* small molecule virulence factors. Microplate assay of **1** (top) and **2** (bottom) against *Pseudonocardia* strains Ae706 and Ae707. *PC is the positive control (50 µg ml$^{-1}$ ciprofloxacin). **NC is the negative control (no inoculation). ***MC, the bacteria grown in LB with 5% of methanol. The pink color indicates viable bacterial cells

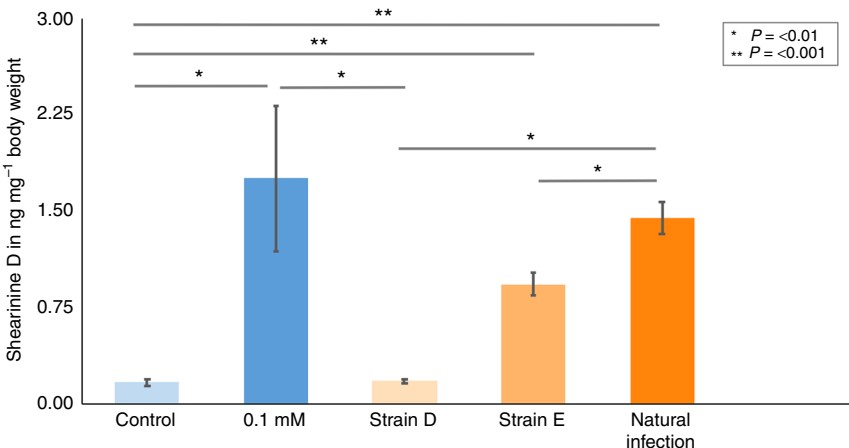

**Fig. 7** Accumulation of virulence factor **2** in *Acromyrmex* ants during nest infection by *Escovopsis*. Levels of **2** detected in leafcutter worker ants (±s.e.m.) reach biologically relevant concentrations during nest infection by *Escovopsis* as follows: *Acromyrmex echinatior* ants fed a 5% glucose solution (control), ants fed a 5% glucose solution containing 0.1 mM **2**, and ants from nests artificially infected with *Escovopsis* strains D or E, as indicated, and *Atta colombica* ants from a nest with a natural outbreak of *Escovopsis* (strain unknown). Lines denote significant differences between groups, as confirmed by Dunn's multiple comparison tests with adjusted *P* values

**Effects of dietary intake of 2 on *A. echinatior* worker ants**. Members of the terpene-indole alkaloid family such as the paspalines[53] and the janthitrems[54–56] are potent mycotoxins and exhibit insecticidal and tremorgenic activity[57]. The structurally related penitrems additionally act as feeding deterrents and modulators of ion channels in various insects[58–60]. The tremorgenic terpene-indole alkaloid **2** has been shown to strongly inhibit the ion channel $BK_{Ca}$, indicating a potent effect on insect nervous systems[37]. The enhanced production levels of **2** during *E. weberi* infection of the fungal cultivar suggested that it could be an important feature in the pathogenesis of *E. weberi*. Even though the pathogen infects the fungal cultivar rather than the ants themselves, we postulated that **2** is a virulence factor primarily affecting the worker ants. Fungal virulence factors are highly diverse but the most well-known example is produced by the fungus *Ophiocordyceps unilateralis*, which causes infected 'zombie' ants to adopt highly unusual behaviors including random walking, repeated convulsions and final death grips underneath leaves[5]. The identity of the virulence factor(s) is not known but transcriptomics data indicate that *O. unilateralis* genes with high similarity to those for the biosynthesis of terpene-indole alkaloids such as penitrems, close structural analogs of the shearinines, are upregulated during manipulated biting behavior of infected carpenter ants[34].

To characterize the effect of **2** on *A. echinatior* ants, we established viable sub-populations of five worker ants in a single Petri dish. These were each supplied with a glucose solution supplemented with concentrations of up to 2 mM of **2** dissolved in 50% methanol (Supplementary Fig. 9 Supplementary Movies 1 and 2) or methanol only as a control. After 10 days, the percentage mortality of the worker ants was significantly affected by the concentration of dietary **2** (Supplementary Fig. 10, Kruskall–Wallis test: $H = 12.6431_{DF=3}$, $P = 0.027$) with greater levels of mortality occurring at higher concentrations. Survival analysis also demonstrated that the probability of survival over time was significantly reduced at increasing concentrations of **2**, even after controlling for different runs of the experiment (Supplementary Fig. 11, Cox's mixed effects model: hazard ratio = 3.83, $z = 0.289$, $P < 0.001$). Supplementation with 50% methanol alone had no effect on ant mortality and all the ants survived the experiment. Additionally, in a control experiment, all ants survived 10 days of exposure to reduced glucose concentrations (dietary concentrations were reduced from 5% to 3% glucose) confirming that mortality was not due to starvation caused by a reduced sugar content at higher concentrations of **2**. In addition to the effects on mortality, we also observed reduced mobility, disorientated movement and loss of balance; ants could no longer successfully traverse the sides and lid of the petri dishes as the level of **2** increased. Time-lapse videos showed that mobility decreased as the concentration of **2** increased, and that each ant spent a longer amount of time sitting stationary on the cotton wool. For example, individual ants in the highest concentration treatment group (2 mM) were stationary for a significantly longer period of time compared to the control treatment group; an average ( ± SE) of 62.8 ± 1.74 s out of the total 65 s of film, compared to an average of 8.8 ± 2.1 s in the control group, respectively ($t_{(7.73)} = 19.74$, $P ≤ 0.001$, Welch's *t*-test) (Supplementary Fig. 12, Supplementary Movies 1 and 2).

To determine the amount of **2** ingested during these experiments, we developed a multiple reaction monitoring (MRM) MS-based method. An authentic reference of **2** was used along with the internal standard yohimbine to scan for selective MS/MS mass transitions. This confirmed that significantly higher concentrations of **2** are present in the worker ant tissues compared to those in control worker ants ($P = 0.001$, Dunn's test) (Fig. 7), and we observed large differences in the tolerated dose of up to 70 ng mg$^{-1}$ of bodyweight (Supplementary Fig. 13). Effects such as unstable movements and reduced mobility were observed in ants from a minimum level of approximately 10 ng of ingested **2** per mg of bodyweight. We also quantified the levels of **2** in worker ants taken from *A. echinatior* sub-colony infection experiments, and from ants in a captive *A. colombica* colony that naturally suffered an uncontrolled outbreak of *Escovopsis* infection (Fig. 7). This confirmed that ants in infected colonies had ingested large amounts of **2**. The levels of **2** in ants during infection with strain E, and also in ants from naturally infected nests, were not significantly different from levels observed in ants that were fed a 5% glucose solution containing 0.1 mM of **2** (Dunn's test: $P = 0.4844$ in both cases respectively) (Fig. 7). Worker ants from control colonies fed 5% glucose solution or from colonies infected with strain D isolated from *T. cornetzi* were indistinguishable (Dunn' test: $P = 0.397$), consistent with the lack of shearinine production by strain D (Fig. 7).

## Discussion

Two major classes of specialized metabolites are produced during infection of the fungal cultivar *L. gongylophorus* by the parasitic fungus *E. weberi* in leafcutter ant nests. Terpene-indole alkaloid **2** is a virulence factor that targets the worker ants and mutualist *Pseudonocardia* bacteria, while ETPs like **1** target the mutualist bacteria only. Combined results from untargeted metabolomics analysis, molecular networking and genome sequencing indicates the occurrence of ETPs and terpene-indole alkaloids in all six *E. weberi* strains isolated from the most highly derived attine genera, *Atta* and *Acromyrmex*, which comprise the leafcutter ants. We conclude that terpene-indole alkaloids play an important role during the phenomenon of fungus garden collapse in leafcutter ant colonies through the manipulation of worker ant behavior. Impairing the ability of *A. echinatior* to efficiently remove infected material by grooming their nests increases the chances of *E. weberi* accelerating chronic low-level infections into acute infestations that will overwhelm entire fungus gardens and allow the parasite to sporulate. This goes some way to explaining previous observations in which fungus gardens are sometimes overwhelmed by *E. weberi*, resulting in the deaths of worker ants before the fungus garden is destroyed. We also hypothesize that both compounds form part of an arsenal in a microbial battle between *E. weberi* and the antifungal-producing *Pseudonocardia* mutualists[20,27–30]. Our study provides direct evidence of an arms race between this specialized mycoparasite and the defensive actinobacterial mutualists of *Acromyrmex* leafcutter ants and their domesticated cultivars.

During the process of review for this manuscript, another study was published where the authors identified that an attine-derived *Escovopsis* strain produces terpene-indole alkaloid shearinines, as well as emodin[61].

## Methods

**Collection of *A. echinatior*.** All ants were sampled from colonies collected during fieldwork in the Gamboa area of the Soberania National Park, Panama (Supplementary Table 1).

**Isolation of *Escovopsis* strains.** For strains originating from *A. echinatior* colonies, *Escovopsis* was isolated from the waste dumps of lab colonies, or by incubating sections of fungus garden, in Petri dishes with moist cotton wool. Strain identity was verified by PCR amplification and sequencing of the 18S gene using the primer set 18S 1A and 18S 564 (Supplementary Table 12). For strains originating from *A. colombica* and *T. cornetzi* colonies, small sections of cultivar fungus were placed on PGA plates under sterile conditions. When viable *Escovopsis* spores were visible, they were harvested and streaked on to PGA plates under sterile conditions, and then restreaked until pure cultures were obtained. Stocks were made from lawns of each purified *Escovopsis* strain grown on 25 ml PGA plates. The spores were harvested by applying 2 ml of sterile glycerol solution (20%, v/v) to the surface of the plate and agitating using sterile cotton buds. The spore suspension from each plate was then transferred to a 2 ml screw cap tubes and stored at −80 °C.

**Cultivation of *E. weberi* and isolation of metabolites.** Cultivation of *E. weberi* strain A was performed by streaking spores onto a PGA plate, allowing confluent growth and then spreading from that plate on to 590 new PGA plates using a damp cotton wool bud. PGA was obtained from Sigma Aldrich (the final composition was 4 g l⁻¹ potato extract, 20 g l⁻¹ dextrose and 15 g l⁻¹ agar). After incubation for 33 days at 20 °C the plates were frozen at −80 °C for 2 h, thawed, and the leaking aqueous phase filtered. Residual agar was extracted with ethyl acetate (2 × 500 ml). The aqueous phase was extracted with ethyl acetate (3 × 250 ml) and all organic fractions were combined. The organic extract was dried over anhydrous sodium sulfate, filtered, and concentrated under reduced pressure. The residue was purified by silica gel open column chromatography (gradient: n-hexane/ethyl acetate: 4/1 to 0/1). Terpene-indole alkaloids and ETPs were present in the 100% ethyl acetate fraction. Semi-preparative HPLC was used for final purification of metabolites using a Gemini® 5 μm NX-C18 110 Å, 150 × 10.0 mm column (Phenomenex). An elution gradient was used starting from H₂O (0.1% formic acid)/MeOH 60/40, to 40/60 within 1 min, to 5/95 within 10 min, to 0/100 within 0.5 min, 0/100 hold for 6.5 min and to 60/40 within 0.5 min.

**Co-cultivation of *E. weberi* strain G and *L. gongylophorus*.** *E. weberi* was used to infect *L. gongylophorus* and compared to the axenic culture in three biological replicates. Initially, a plug of 8 mm of *L. gongylophorus* fungus garden was placed on the edge of a 9 cm diameter petri dish filled with 25 ml of PGA and grown for 5 days at 25 °C. An 8 mm diameter plug taken from a confluent PGA plate of *E. weberi* strain G was then added on the center of the plate and cocultured with *L. gongylophorus* for 5 days under the same conditions. The agar was extracted with ethyl acetate (25 ml) and dried with anhydrous sodium sulfate. After filtration, the solvent was removed under reduced pressure from an aliquot (4 ml). The residue was dissolved in methanol (1 ml) for further analysis. Aliquots (15 μl) were analyzed for the content of **1** and **2** as described below.

Quantitative analysis was performed using a Shimadzu Prominence system with LC-20AT pumps, a SIL-20ACHT autosampler and a SPD-M20A PDA detector. Chromatography was achieved using a Nucleodur® C18 HTec, 5 μm, 110 Å, 150 × 10.0 mm column (Macherey-Nagel). A gradient was used starting from H₂O (0.1% formic acid)/ACN 99.5/0.5 within 1 min, to 0/100 within 35 min, 0/100 hold for 5 min, and to 99.5/0.5 within 1 min. **1** and **2** showed retention times of 23.2 min and 28.7 min respectively. Quantitative analysis was performed by integrating the areas under the peaks (at 254 nm). Mean area of **1** for the co-culture ($n = 3$; 34156 ± 4839 au²) and for the axenic culture ($n = 3$; 10120 ± 4052 au²); mean area of **2** for the co-culture ($n = 3$; 10504 ± 1798 au²) and for the axenic culture ($n = 3$; 1175 ± 182 au²). This result confirmed that from a similar experiment which had been run previously and independently in Norwich and which gave similar results (Supplementary Fig. 1).

**Spectroscopy.** NMR was performed on a Bruker AVANCE III 400 MHz spectrometer. Chemical shifts are reported in parts per million (ppm) relative to the solvent residual peak of chloroform-d₁ (¹H: 7.24 ppm, singlet; ¹³C: 77.00 ppm, triplet) or DMSO-d₆ (¹H: 2.50 ppm, quintet; ¹³C: 39.52 ppm, septet). The Specific optical rotation was measured with a Model 341 Polarimeter (PerkinElmer Inc.). A Lambda 35 UV/Vis spectrometer (PerkinElmer) was used for ultraviolet/visible (UV/Vis) spectroscopy. Unless otherwise mentioned, UPLC-MS measurements were performed on a Nexera X2 liquid chromatograph (LC-30AD) system (Shimadzu) connected to an autosampler (SIL-30AC), a Prominence column oven (CTO-20AC) and a Prominence photo diode array detector (SPD-M20A). The UPLC System was connected to a LCMS-IT-TOF Liquid Chromatograph mass spectrometer (Shimadzu). A Kinetex® 1.7 μm C18 100 Å, 100 × 2.1 mm column (Phenomenex) was used for chromatographic separation and the column oven temperature was set to 30 °C. The mobile phase was a mixture of solvent A (0.1% formic acid in water) and solvent B (methanol) with a gradient as follows: solvent A/B initial condition 90/10, hold at 90/10 for 1 min, linear gradient up to 0/100 within 9.00 min, hold for 2.00 min, returned to 90/10 within 0.5 min, hold at 90/10 for 0.5 min. MS spectra were acquired within a mass range of *m/z* 170–1700 using an ion accumulation time of 20 ms per spectrum. We used the following parameters for MS analysis: temperature of the curved desolvation line 250 °C; temperature of the heat block 300 °C; nebulizer gas flow 1.5 l min⁻¹; interface (probe) voltage −3.5 kV for negative mode and 4.5 kV for positive mode. The detector voltage of the time-of-flight (TOF) mass analyzer was set to 1.66 kV and the collision-induced dissociation energy to 50%. The instrument was calibrated using sodium trifluoroacetate cluster ions according to the manufacturer's instructions. Spectra are shown in Supplementary Figs. 14-31.

**Genome sequencing and assembly.** Lawns of *Escovopsis* were grown on top of sterilized cellophane discs on PGA medium. After 2 weeks of incubation at room temperature, the fungal mycelium was scrapped off into sterile tubes. Mycelial material was crushed in a pestle and mortar with liquid N₂. Crushed freeze-dried material was then used with the QIAGEN DNeasy Plant mini kit to isolate DNA. Illumina sequencing of DNA was carried out at the DNA Sequencing Facility, Department of Biochemistry, University of Cambridge, UK, using TruSeq PCR-free and Nextera Mate Pair libraries and a MiSeq 600 sequencer. Genome assembly was performed using Roche Newbler v3.0, scaffolds were polished using PILON version 1.13, and reads were mapped using Burrows–Wheeler transformation version 0.7.12-r1039. BGCs were identified using BLAST 2.2.31+ with amino acid or nucleotide sequences from the published, annotated BGCs.

**Sample preparation for MS analysis.** *Escovopsis* strains were grown on PGA medium in five biological replicates for 30 days at 20 °C. Mycelium and agar was disintegrated, transferred into a beaker and extracted with a 1:1 mixture of acetonitrile and methanol (15 ml). The supernatant was filtered, evaporated and the residue dissolved in methanol. Samples were analyzed as described in the Spectroscopy section above.

**Molecular networking analysis.** UPLC-HRMS chemical profiling data obtained from the Shimadzu IT-TOF system were converted to the mzXML format, uploaded to the GNPS server with FileZilla 3.25.1, and processed using the GNPS web platform. Cytoscape 3.4.0 was used to analyze, organize and visualize data. For the molecular networking analysis we used the following parameters: precursor ion mass tolerance 2.0 Da; fragment ion mass tolerance 0.5 Da; advanced network option: pairs min cosine: 0.5; minimum matched fragment Ions: 2; minimum cluster size: 1; network TopK: 50; maximum connected component size (Beta): 100; advanced library search options: library search min matched peaks: 2; score

threshold: 0.7; analog search disabled; maximum analog search mass difference: 100.0; advanced filtering options: filter below Std Dev: 0.0; minimum peak intensity: 0.0; filter precursor window, filter peaks in 50 Da window and filter library.

**MIC determination**. Stock solutions of **1** and **2** were prepared in methanol and further diluted in Luria-Bertani (LB) broth to give concentration in the range 1.5 ng ml$^{-1}$ to 100 µg ml$^{-1}$. Aliquots of each solution (100 µl) were transferred to a 96-well microplate followed by the inoculation of 5 µl of a 1:10 (v/v) 0.5 McFarland suspension of two different *Pseudonocardia* mutualist strains isolated previously from *A. echinatior* colonies collected in Gamboa, Panama[27]. The final concentration of the bacteria was approximately $5 \times 10^4$ colony-forming units per ml per well. After 20 h of incubation time at 30 °C and 150 rpm, 5 µl of resazurin solution (6.75 mg ml$^{-1}$ in deionized water) was added and the color changes were observed prior to 4 h of incubation to determine the MIC. Ciprofloxacin was used as a positive control. To exclude an inhibitory effect of the solvent, control experiments with methanol in LB were performed. All the assays were performed in triplicate.

**Dietary supplementation with 2**. Purified **2** was dissolved in 50% methanol to give a stock concentration of 5 mM which was diluted with a 5% glucose solution to concentrations of 0.1 mM, 0.25 mM, 0.5 mM, 1 mM and 2 mM, respectively. Individual *A. echinatior* worker ants were removed from colonies and placed in 9 cm petri dishes containing a $3 \times 2$ cm piece of cotton wool soaked in water to maintain humidity. Groups of five ants were supplied with one of the five different concentrations of **2** in 5% glucose (300 µl) or methanol only in 5% glucose (300 µl) as a control for 10 days. In a separate control experiment, ants were exposed to 300 µl of 5, 4.9, 4.75, 4.5, 4 and 3% glucose solutions for 10 days. These reduced concentrations corresponded to the reduced amount of glucose in treatments receiving higher concentrations of compound **2** in the first experiment.

Solutions were supplied to ants in an Eppendorf lid. Water was added to the cotton wool daily and Eppendorf caps were topped up with the respective treatment every 3–4 days. Dead ants in each concentration treatment were scored daily, collected and frozen. Differences in individual motility and the occurrence of unusual movements were monitored by eye via time-lapse videos, which ran over 3 h with 10 s intervals between frames (this amounted to approximately 1 min of video). The amount of time spent stationary by each individual ant was timed (in seconds) between the highest concentration treatment group (2 mM) and the control group, over the course of each time-lapse video. All ants were collected at their time of death or at the end of the 10 days. Samples were immediately frozen at −80 °C and stored until further analysis.

**Experimental *Escovopsis* sub-colony infections**. Two separate sub-colonies of *A. echinatior* were created by taking pieces of fungus garden, measuring approximately 6 cm$^3$, from captive colony Ae088. For each sub-colony, the fungus was placed in a plastic container inside a larger plastic box. Each fungus piece was covered in $20 \times 3$ µl drops of *Escovopsis* spore suspension. Spores were prepared by growing confluent *Escovopsis* lawns on PGA plates and removing spores with a cotton bud and 500 µl of sterile water. Each sub-colony received a different strain of *Escovopsis*, either strain D (isolated from a *Trachymyrmex cornetzi* nest) or strain E (isolated from an *Acromyrmex echinatior* nest). The same number (25 of each caste) of forager and fungus garden worker ants were placed into each sub-colony. The ants were additionally supplied with bramble leaves and cotton wool soaked in water. After 2 weeks, samples of ants, fungus garden and waste dump were taken from sub-colonies and used for chemical analysis.

**MS-based quantitation of 1 and 2**. Individual *A. echinatior* ants and samples of *L. gongylophorus* were stored at −80 °C immediately after collection. To prepare samples for testing, they were warmed to room temperature and their weight determined by a high precision balance. Ants were washed in methanol (for 5 s) to remove potential contamination with **2** on their outer surface. Subsequently, each sample was transferred to a centrifuge tube (15 ml; Corning Incorporated), frozen in liquid nitrogen and mechanically disintegrated by the back end of a disposable spreader. The residue was extracted with 980 µl of methanol. The internal standard, yohimbine (20 µl of a 10 nM solution in methanol), was added. Samples were thoroughly mixed, centrifuged, the supernatant removed, evaporated under reduced pressure and stored at −80 °C. The samples were redissolved in methanol (500 µl) prior to analysis.

Spectra were acquired on a Xevo TQS tandem quadrupole mass spectrometer. We developed a MRM MS method for the quantitation of shearinine natural products featuring at a capillary voltage of 2.00 V; source temperature of 150 °C; desolvation temperature of 500 °C; desolvation gas flow of 800 l h$^{-1}$; cone gas flow of 150 l h$^{-1}$; nebulizer pressure of 7 bar; and a collision gas flow of 0.15 ml min$^{-1}$. For analysis of **2** we optimized the cone voltage (to 30 V) and the collision energies for each of the four transitions (Supplementary Table 13). During parameter optimization, we continuously injected a solution of **2** (0.1 µM in methanol) into a flow of 0.4 ml min$^{-1}$ of 65% acetonitrile (eluting conditions of the analyte). We used the mass transition 600.4 > 238.2 for quantitation.

For quantitation of **1** we optimized the cone voltage (to 36 V) and the collision energies for each transition (Supplementary Table 13). During parameter optimization, we continuously injected a solution of **1** into a flow of 0.4 ml min$^{-1}$

of 65% acetonitrile (eluting conditions of **1**) into the electrospray ionization source. We used the mass transition 729.1 > 282.1 for quantitation. For the internal standard Yohimbine we again optimized the cone voltage (to 98 V) and the collision energies for each transition (Supplementary Table 13). During parameter optimization, we continuously injected a solution of Yohimbine into a flow of 0.4 ml min$^{-1}$ of 35% acetonitrile (eluting conditions of the internal standard). We used the mass transition 355.3 > 212.2 (22 V) for quantitation.

Metabolite separation was achieved using an ACQUITY UPLC system (Waters) system and a Kinetex® 1.7 µm C18 100 Å, $50 \times 2.1$ mm column (Phenomenex). Chromatographic conditions were as follows: starting conditions, H$_2$O (0.1% formic acid)/acetonitrile: 80/20, hold for 0.3 min, up to 30/70 within 1.7 min, up to 0/100 within 1 min, hold for 1.3 min, to 80/20 within 0.1 min, hold for 0.6 min. The total run time was 5 min. A flow rate of 0.4 ml min$^{-1}$ and a column temperature of 40 °C was used. For each analysis 2 µl of sample was injected using partial-loop needle overfill (PLNO) mode. The weak wash consisted of 500 µl of 10% acetonitrile and 90% water (containing 0.01% formic acid) and the strong wash solution 500 µl consisted of 100% acetonitrile (containing 0.01% formic acid).

Calibration standards were prepared by diluting a stock solution of **1** or **2** (1 mM in methanol) to give calibrants with concentrations of 1 nM, 5 nM, 10 nM, 25 nM, 50 nM, 75 nM, 100 nM, 250 nM, 500 nM, 750 nM and 1 µM. The calibration curve was generated by injecting 2 µl of each standard (PLNO) directly after analysis of the respective samples of each day.

**Data availability**. The authors declare that the data supporting the findings reported in this study are available within the article and the Supplementary Information, or are available from the authors on reasonable request. New nucleotide sequence data have been deposited in NCBI GenBank under the accession codes as follows: *E. weberi* strain A (NIGB00000000); *E. weberi* strain B (NQYR00000000); *E. weberi* strain C (NQYS00000000); *Escovopsis* strain D (NIGD00000000); *E. weberi* strain E (NQYQ00000000); *E. weberi* strain F (NIGC00000000). Raw reads have been deposited at the NCBI Sequencing Reads Archive with accession codes as follows: *E. weberi* strain A (SRP117677); *E. weberi* strain B (SRP120297); *E. weberi* strain C (SRP120188); *Escovopsis* strain D (SRP136533); *E. weberi* strain E (SRP122917); *E. weberi* strain F (SRP117700). Nucleotide sequences and annotations of the BGCs identified in *E. weberi* strain G are available in the Third Party Annotation Section of the DDBJ/ENA/GenBank databases under the accession numbers TPA: BK010418-BK010420. The ITS sequence for *Escovopsis* strain D was amplified with ITS1 and ITS4, sequenced and deposited with Genbank accession: MG897412.

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

## Acknowledgements

This work was supported by research grants NE/M015033/1 and NE/M014657/1 from the Natural Environment Research Council (NERC) to M.I.H., B.W., J.C.M. and D.W.Y.; by the Biotechnology and Bioscience Research Council (BBSRC) via Institute Strategic Programme Grants BB/J004561/1 and BB/P012523/1 to the John Innes Centre; by the DFG (ChemBioSys, CRC 1127, and Leibniz Prize) to C.H.; and by an Advanced ERC Grant (323085) to J.J.B. which also funded T.M.I., along with a Marie Curie Individual European Fellowship (IEF grant 627949). S.F.W. was funded by a NERC PhD student-ship (NERC Doctoral Training Progamme grant NE/L002582/1). The Smithsonian Tropical Research Institute provided logistical help and facilities to work in Gamboa, and the Autoridad Nacional del Ambiente yel Mar gave permission to sample and export ants from Panama. We thank the DNA Sequencing Facility, Department of Biochemistry, University of Cambridge UK, especially Dr. Markiyan Samborskyy, for sequencing and assembling the *Escovopsis* genomes. We also thank Dr. Lionel Hill and Dr. Gerhard Saalbach (Molecular Analysis Services, JIC) for their excellent metabolomics support.

## Author contributions

B.W., C.H., P.C.V., D.H., D.W.Y., J.C.M., J.J.B. and M.I.H. designed the research. B.W., D.H., J.J.B., M.I.H., N.A.H. and S.F.W. wrote the manuscript and all authors commented. S.F.W. performed the ant behavioral experiments. T.M.I. isolated the *Escovopsis* strains B-F from ant waste dumps of colonies maintained in Copenhagen. N.A.H. performed the molecular microbiology experiments, phylogenetics and genome analysis. E.P. maintained captive *Atta* and *Acromyrmex* colonies in Norwich. D.H., A.C.A.S. and K.S. isolated metabolites, performed chemical analysis and MIC determination.

## Additional information

**Competing interests:** The authors declare no competing interests.

