## [Peer Review File · Nature Communications]

Reviewers' comments:

Reviewer #1 (Remarks to the Author):

The authors have investigated the antagonistic effect of *Escovopsis* fungi on the symbiotic organisms *Leucoagaricus*, the fungus cultivated by *Acromyrmex* leafcutter ants, *Acromyrmex* itself, and *Pseudonocardia*, the mutualistic bacteria that provide these organisms with antibiotics. They focused their research on the identification of natural compounds produced by *Escovopsis* that might play a major role in *Acromyrmex* colony collapse. Using a multi-faceted approach, the authors have identified the first *E. weberi* virulence factors. This molecular identification is an exciting endeavor and highly relevant to a wider field.

While I think the work presented here is generally exciting and convincing, I am of the opinion that some additional controls are needed with regard to the dietary supplementation experiments and the effect of Shearinine D on ant behavior. Shearinine D is dissolved in 50% methanol, after which the stock is further diluted with sugar water to be fed to the ants. This means that in the highest concentrations of Shearinine D (1 mM and 2 mM, respectively) 10 % and 20 % methanol is present. In addition, sugar content is lower than the 5 % sugar water fed to the controls (4 % and 3 %, respectively). To account for this, additional controls should be set up in which ants are exposed to similar concentrations of methanol and sugar without the Shearinine D to assure that the changes in behavior observed are indeed an effect of the bioactive compound and not an effect of the methanol or starvation (the latter already being given as an option since Shearinine might work as a deterrent).

My concern is of similar nature with regards to the MIC determination if compound 2 was dissolved in methanol prior to preparation in LB. If this was the case, additional controls are needed to determine the true effect of the bioactive compound on bacterial growth vs the possible effect of methanol. Dissolving in methanol is however not mentioned in the materials and methods for these experiments and might not have been necessary for the concentrations used here.

Regarding the claim that Shearinine D has a behavioral effect in *Acromyrmex*, I would also like to request additional video material that shows the penitrem-like effect in these ants. If Shearinine D works like a penitrem, and the lethargic effect that has been recorded is not a mere effect of methanol, observation of tremors, i.e. rhythmic twitching of the legs, might be expected. The supplementary videos currently do not provide this detail. This detail is, in my opinion, needed to make any real claim towards behavioral manipulation beyond mere lethargy. Are tremors additionally observed in the naturally infected ants in which increased levels of Shearinine D have been measured? Taken together, these observations would provide a more confident claim for behavioral manipulation.

The authors rightfully mention that ants exposed to Shearinine D in the dietary supplementation experiments might be starving because they are deterred by the compounds presence in the sugar water. This reviewer wonders if this could be tested by determining the exact volume or weight before and after introducing the eppendorf cap with the diet. Instead of topping up every 3 days, the cap could be exchanged for a new one. By including controls for the evaporation rate of the different dietary compositions (higher methanol concentrations might evaporate faster), the authors could also determine how concentrations of Shearinine D might be increasing over time due to possible evaporation. The current experiments do not control for this.

Minor comments:

For the experimental *E. weberi* sub-colony infections, the spore concentration of the spore suspension used is not given. This should be provided to facilitate possible replication by others.

In the introduction's last paragraph: the authors should only refer to reference 34 for the discussion of involvement of penitrem-like compounds in zombie ant biting behavior. These compounds are not discussed in reference 4.

Was signed,

Charissa de Bekker

Reviewer #2 (Remarks to the Author):

This manuscript by Heine and Holmes, et al. addresses the identification and characterization of two secondary metabolites from *Escovopsis* that directly and indirectly affect the survival of *Acromyrmex* leaf-cutter ants. The authors combine methods from various fields to elucidate the function of these two secondary metabolites. This is the first time these compounds have been shown to have the specific biological activities outlined in the manuscript. The manuscript focuses heavily on the chemistry used, but the biological and ecological function of these compounds is really the exciting part of this work and likely to be of interest to the wider community. While I am enthusiastic about this topic and think it is appropriate for publication in *Nature Communications*, I do not feel that this manuscript is ready for publication in its present form. Overall, I think this manuscript could be improved by thorough revision by the authors with streamlined formatting. In particular, I found the manuscript difficult to read and interpret as the information was largely repetitive and written more as a catalog of experiments than as a final polished paper.

Major concerns:

1. The formatting of the manuscript makes the underlying story rather difficult to follow. The "Results" section is really a Results and Discussion section and the "Discussion" is actually the Conclusion. In addition, the untargeted metabolomics analysis is discussed after the targeted structural elucidation of two "major classes of secondary metabolites". I understand organizing the manuscript in this way. However, considering that the two compounds isolated were the ones tested for biological activity, which is discussed last, the untargeted analysis is rather repetitive and orthogonal to the main work in the paper. If the authors are set on including all of the data in a single manuscript, I suggest discussing the untargeted approach first, using the interaction to justify focusing on the two main compounds, then discussing the biological activity of the two compounds. Since the focus of the paper is actually the biological activity, it may not be necessary to discuss the full structural isolation and NMR elucidation of the two main compounds as they have been previously described. Further discussion of the BGCs and/or biological activity of the compounds would enhance the manuscript.
2. Data is not consistently quantified in the manuscript. For example, Figure 2 shows the absorbance profiles of the extracts of the interaction of *L. gongylophous* and *E. weberi* plus their control axenic cultures. In the text, the authors state "Both 1 & 2 were absent in extracts of the healthy food fungus, but were elevated approx. three-fold when compared to axenic cultures of *E. weberi* (Fig. 2)." Based upon the figure presented, I cannot draw that conclusion. This data can and should be quantified by using area under the curve values. Further, in the methods section the statement "Quantitative analysis was performed..." indicates that these values were calculated, although not included. Another example is Figure 5. The MICs of the two main compounds against two strains of *Pseudonocardia* were measured via resazurin. However, the figure displays a photograph of a single replicate of the MIC measurements. Visually, even via pdf on a monitor, it is hard to distinguish the pink/purple/blue colors of the photograph. Resazurin conversion to resorufin can be quantified using fluorescence (or absorbance) measurement.
3. It would be helpful if the authors included additional data in their supplemental files. Inclusion of NMR spectra (versus the listed ppm shifts) and MS data (MS2 spectra, MRM quantitation, etc.) would

provide stronger support of the structural conclusions and quantification measurements. I request this because the settings for molecular networking are quite broad for high-resolution mass spectrometry data. The precursor ion mass tolerance of 2.0 Da and the fragment ion mass tolerance of 0.5 Da are the default settings for molecular networking. This means that the precursor ion mass window is 4.0 Da. Typically for HRMS data, assuming 10 ppm error, a precursor ion mass tolerance of 0.05 Da (precursor ion mass window of 0.1 Da) should be sufficient across the entire measured mass range. In addition, the cosine score is set to 0.5 and the minimum matched paired peaks is set to two. The cosine score is rather low (default setting is 0.7), but the concern is really the minimum matched paired peaks. This means that in order for two MS/MS spectra to show similarity, the MS/MS spectra only have to contain two overlapping or related peaks. These settings may have been required in order to ensure that the compounds of interest networked, but the MS/MS spectra should be included in the supplemental information as lower settings for cosine and minimum matched paired peaks are more likely to allow unrelated MS/MS spectra to cluster. Inclusion of the MS² spectra would allow validation of the networking results.

4. Figure 4 is illegible. It may make more sense to move the large heatmap and molecular network to the supplemental information. Supplemental Table 4 would provide a better visualization of the heatmap and the sub-network of the melinacidin/chetracin molecular family shown in Supplemental Figure 5 would be more useful in the main text. It would be helpful if the authors labeled the cosine scores between two connected nodes for the enlarged figures of the molecular families. It may be beneficial if the sub-network for the terpenoid indole alkaloids was also included.

Minor concerns:

General:

The authors use standard error for their statistical calculations, which suggests that they aim to compare the averages between the different experimental groups. However, very few of these comparisons are made in the text or denoted in the bar graphs.

ETPs is used throughout, but it is not defined.

ppm error should be reported for (putatively) annotated compounds.

There is a lot of gene information included in the manuscript, but unfortunately, is not discussed at length.

Gene numbers should be included in genome/BGC annotations, if available.

Supplementary Figure 2 is missing some of the NOESY data (arrows without shifts).

Supplementary Figure 11 appears to have had conversion issues.

It is unclear why yohimbine was used as an internal standard for quantification. Was this because it is also an indole alkaloid?

Introduction:

Paragraph 1

Lines 6-7: "...immunity of their colonies. Well known examples are the..."

Unclear. I am assuming the examples are eluding to fungi that specialize on ants as host.

Paragraph 2

Lines 7-9: "The gongylidia serve as the sole...fresh leaf pulp."

This sentence does not make sense. May want to consider rewording as "The gongylidia have two functions: the sole food source for ant larvae and as a source of enzymes for leaf pulp decomposition." Or something similar.

Paragraph 3: This paragraph has abrupt changes between commensal and invasive microbes. May want to consider starting a new paragraph at "It has been estimated..."

Paragraph 3

Lines 12-14: "In *A. echinator*, such events reflect a failure..."

This sentence should be simplified.

Lines 14-15: "The colony collapse that typically follows..."

Follows what event?

Paragraph 4

Line 1: secondary or specialized is sufficient.

Line 3-5: simplify by removing both usages of "both"

Line 10: E. weberi should be spelled out for first use.

Results:

Paragraph 1

Line 3: "...and comparing to axenic cultures..."

Comparing what features of the co-culture experiment to the axenic cultures?

Lines 5-6: "...of two major and a range of minor metabolites..."

What defines major and minor? Amount produced? Bioactivity?

Line 6-7: "...had signals of m/z = 729.0937..."

Change "of" to "at" and remove "="

Line 9: "Essentially identical..."

What does this mean? How can this conclusion be drawn when the gradients were clearly different and the fold change is 7 versus 3.

Paragraph 2

Lines 1-3: This sentence does not make sense. Presumably, a large scale cultivation of the axenic E. weberi was performed to isolate the compounds.

Line 7: Replace "nature" with "compound".

Paragraph 3:

Line 15: Reference is needed after "...same Panamanian field site."

Paragraph 4:

Line 4: Remove "that we"

Line 7: Change to "UPLC-MS/MS" (may need to define MS/MS)

Line 19: Remove "finally"

Paragraph 5

Lines 1-2: Remove "that we aimed to unambiguously identify."

Line 2: "Production levels were sufficiently..."

Production levels of what compounds?

Paragraph 6

Line 5: "The metabolomics data clearly showed..."

The heatmap associated with this conclusion is not clear. See major concern #4.

Paragraph 7

Lines 11-12: Remove "fungal antiSMASH"; Add "from fungus" after BGCs.

Line 15: Removed "could"; change tense of "identify" to "identified"

Paragraph 9

Line 10-13: "...famous example is produced..."

This sentence is incomplete. What is the example fungal virulence factor?

Paragraph 10

Line 3: Remove which could be.

Paragraph 11

Line 8-9: Reference is needed after "...rather than poisoning."

Line 9: Remove "then"

Discussion

Paragraph 1

Line 6: Remove "widespread"

Materials and Methods

Paragraph 1: Were appropriate permits gathered (if necessary)?

Paragraph 2

Line 3: To what does "them" refer?

Line 3-4: Change "amplify" to "amplification"; What gene was used?

Line 8-10: This sentence does not make sense and would not be reproducible.

Paragraph 4: Considering that two experiments measuring the two major metabolites were performed in different labs under clearly different gradients (figure 2 and suppl. figure 1), there should be two sets of HPLC methods.

Paragraph 5: For the HRMS, was internal and/or external calibration used? What was the eV setting used for CID?

Paragraph 7

Lines 5-6: "Samples were analyzed the standard method described in Spectroscopy section above."
What does this mean?

Paragraph 8

Line 2: Change "transferred" to "converted"

Paragraph 10: It is not clear how the movement of each ant was quantified. Was this analysis performed manually or through a program?

Line 2: Remove "then"

Line 15: Remove "at the latest"

Paragraph 12: For the MRM method, what mass spectrometer was used as the UPLC system is Waters not Shimadzu.

Line 4: Remove "additionally"

Reviewer #3 (Remarks to the Author):

The manuscript provides new insight into the chemical warfare of leafcutter ant symbionts and pathogenic Escovopsis. In the manuscript, the authors present direct genomic and metabolomic evidence describing the virulence factors produced by invading fungus, and evaluate their activity on the mutualistic *Pseudonocardia* and host ant. The authors complete a comprehensive study identifying, and quantifying these compounds and their congeners, and point to a genomic origin for these compounds. However, their genomic analysis is less comprehensive with insufficient evidence presented to objectively evaluate their arguments. Moreover, this analysis is peripheral to the main conclusions of the work with no justification or deeper analysis. Overall, the manuscript is well written with few errors and will provide valuable information to entomologists working with leafcutter ants.

Specific questions

It is unclear how strong the BCG evidence is as the level of homology is not discussed. How similar are these clusters to each other and the known/characterized cluster? How do you assess the accuracy of your characterization e.g. EmoH == decarboxylases?

Do the differences in clusters explain the differences in congener profiles? I.e. More divergent gene clusters preferentially produce another congener?

It is unclear what Figure 4A is depicting both in image resolution and data representation. What metabolites do the rows represent? What do the columns represent? At first glance, these look like they represent different strains of *E. weberi* but these strains have each been subdivided into distinct columns. How was the data organized? why are the shearinines not together, for example?

Supp Fig 7 – Strains E, F, G mislabeled

Why was *tef1* alone used for phylogenetic analysis of *Escovopsis* as opposed to the more common ITS1/ITS2 standard for fungal species? See Schoch, C. L. et al. Nuclear ribosomal internal transcribed spacer (ITS) region as a universal DNA barcode marker for Fungi. *PNAS* 109, 6241–6246 (2012); Meirelles LA, Montoya QV, Solomon SE, Rodrigues A (2015) New Light on the Systematics of Fungi

Associated with Attine Ant Gardens and the Description of *Escovopsis kreiselii* sp. nov. PLoS ONE 10(1): e0112067, ref 49

"with five species recognized" change to "with five classified species"

How does your clade analysis align with that in ref 49 as your phylogenetic analysis use different reference genes? Also, ref 49 defines no *E. weberi* species as a member of the 9 proposed clades (para. On Genome sequencing)

There are broken characters in Sup Fig 11 (boxes).

How do you determine that the species you have isolated are actually *E. weberi* vs and *Escovopsis* with 'weberi morphology' as described in ref 49?

Reviewer #1 (Remarks to the Author):

The authors have investigated the antagonistic effect of *Escovopsis* fungi on the symbiotic organisms *Leucoagaricus*, the fungus cultivated by *Acromyrmex* leafcutter ants, *Acromyrmex* itself, and *Pseudonocardia*, the mutualistic bacteria that provide these organisms with antibiotics. They focused their research on the identification of natural compounds produced by *Escovopsis* that might play a major role in *Acromyrmex* colony collapse. Using a multi-faceted approach, the authors have identified the first *E. weberi* virulence factors. This molecular identification is an exciting endeavor and highly relevant to a wider field.

While I think the work presented here is generally exciting and convincing, I am of the opinion that some additional controls are needed with regard to the dietary supplementation experiments and the effect of Shearinine D on ant behavior. Shearinine D is dissolved in 50% methanol, after which the stock is further diluted with sugar water to be fed to the ants. This means that in the highest concentrations of Shearinine D (1 mM and 2 mM, respectively) 10 % and 20 % methanol is present. In addition, sugar content is lower than the 5 % sugar water fed to the controls (4 % and 3 %, respectively). To account for this, additional controls should be set up in which ants are exposed to similar concentrations of methanol and sugar without the Shearinine D to assure that the changes in behavior observed are indeed an effect of the bioactive compound and not an effect of the methanol or starvation (the latter already being given as an option since Shearinine might work as a deterrent).

>All the experiments were run in parallel using methanol-only (including 10% and 20% methanol) controls and these had no obvious effect on mortality; all the ants survived the 10-day experiment. We have added this information to the appropriate methods and results sections in the manuscript. The reviewer is correct about the drop in the glucose concentration (to 4% and 3%, respectively). Previous experiments have shown that ants survive and behave normally for ≥ 10 days with no glucose as long as water is available, so we do not believe reducing the glucose concentration would have any effect on ant mortality or behaviour. However, we have run the control experiment whereby ants were exposed to dietary solutions containing a lower concentration of glucose with no supplementation. Groups of 5 ants were given 5%, 4.9%, 4.75%, 4.5%, 4% and 3% glucose solution, respectively; these concentrations correspond with the reduced amount of sugar that occurred during the shearinine experiment. All ants survived the 10-day experiment.

My concern is of similar nature with regards to the MIC determination if compound 2 was dissolved in methanol prior to preparation in LB. If this was the case, additional controls are needed to determine the true effect of the bioactive compound on bacterial growth vs the possible effect of methanol. Dissolving in methanol is however not mentioned in the materials and methods for these experiments and might not have been necessary for the concentrations used here.

> Both compounds (1 and 2) were dissolved in methanol prior to dilution in LB. Stock solutions of 2 mg/mL in methanol were prepared and then further diluted in LB. Thus, the solution with the highest concentration (100 μ g/mL) also contained a higher volume of methanol (5%), however, appropriate

control experiments with MeOH in LB were performed to exclude any effect from the solvent. We added the missing information to the materials and methods section and updated **Fig. 5**.

Regarding the claim that Shearinine D has a behavioral effect in *Acromyrmex*, I would also like to request additional video material that shows the penitrem-like effect in these ants. If Shearinine D works like a penitrem, and the lethargic effect that has been recorded is not a mere effect of methanol, observation of tremors, i.e. rhythmic twitching of the legs, might be expected. The supplementary videos currently do not provide this detail. This detail is, in my opinion, needed to make any real claim towards behavioral manipulation beyond mere lethargy. Are tremors additionally observed in the naturally infected ants in which increased levels of Shearinine D have been measured? Taken together, these observations would provide a more confident claim for behavioral manipulation.

>Methanol alone has no effect on ant behaviour or mortality (see above). We accept that ‘tremors’ was a poor choice of word, we meant to convey that worker ants treated with shearinine D are unstable and do not balance well, e.g. they have trouble climbing and fall off the walls and lids of the petri dishes, as can be observed on SI videos 1 and 2. We have made this clearer in the text. Sadly, we cannot record the ants in the naturally infected colony because it died in the process; we were, however, able to use this material for post-mortem analysis of shearinine D concentration.

The authors rightfully mention that ants exposed to Shearinine D in the dietary supplementation experiments might be starving because they are deterred by the compounds presence in the sugar water. This reviewer wonders if this could be tested by determining the exact volume or weight before and after introducing the eppendorf cap with the diet. Instead of topping up every 3 days, the cap could be exchanged for a new one. By including controls for the evaporation rate of the different dietary compositions (higher methanol concentrations might evaporate faster), the authors could also determine how concentrations of Shearinine D might be increasing over time due to possible evaporation. The current experiments do not control for this.

> We do not think this is technically feasible given the tiny weights involved. However, our data show that the ants are consuming shearinine from the sugar water because it accumulates in the ant tissues. We have also shown that higher levels of shearinine D in sugar water accumulate at higher levels in the ant tissues and are more toxic. Also, the ants survive for the whole length of the experiment without any sugar water as long as water is provided (see above) so they are not starving. The reviewers all agree that these data are convincing.

Minor comments:

For the experimental *E. weberi* sub-colony infections, the spore concentration of the spore suspension used is not given. This should be provided to facilitate possible replication by others.

>We did not measure the spore concentration since we find titres of spore preparations to be very unreliable after storing them for any length of time. Instead we set up a method that allowed us to consistently use the same fresh inoculum and we have added the following sentence to the methods so that others can replicate these experiments: “*E. weberi* spores were prepared by harvesting a 2 cm² square of a confluent 25 ml PGA plate of *E. weberi* using a cotton bud and 1ml of 20% glycerol and then placing 10 x 2µl of this spore suspension onto 2cm³ of the *L. gongylophorus* nest for five days under the same conditions.”

In the introduction’s last paragraph: the authors should only refer to reference 34 for the discussion of involvement of penitrem-like compounds in zombie ant biting behavior. These compounds are not discussed in reference 4.

>This citation has been removed

Reviewer #2 (Remarks to the Author):

This manuscript by Heine and Holmes, et al. addresses the identification and characterization of two secondary metabolites from *Escovopsis* that directly and indirectly affect the survival of *Acromyrmex* leaf-cutter ants. The authors combine methods from various fields to elucidate the function of these two secondary metabolites. This is the first time these compounds have been shown to have the specific biological activities outlined in the manuscript. The manuscript focuses heavily on the chemistry used, but the biological and ecological function of these compounds is really the exciting part of this work and likely to be of interest to the wider community. While I am enthusiastic about this topic and think it is appropriate for publication in Nature Communications, I do not feel that this manuscript is ready for publication in its present form. Overall, I think this manuscript could be improved by thorough revision by the authors with streamlined formatting. In particular, I found the manuscript difficult to read and interpret as the information was largely repetitive and written more as a catalog of experiments than as a final polished paper.

Major concerns:

1. The formatting of the manuscript makes the underlying story rather difficult to follow. The “Results” section is really a Results and Discussion section and the “Discussion” is actually the Conclusion.

>The other two reviewers are happy with the way the manuscript is written so we are reluctant to change the order. Indeed, reviewer 1 states ‘I think the work presented here is generally exciting and convincing...’ and reviewer 3 states ‘Overall, the manuscript is well written with few errors...’ We would prefer not to change the order in which the results are presented. However, we agree that, occasionally, excessive detail may have been included in the text, and have therefore made best efforts to streamline the narrative by removing details into the supplementary information wherever appropriate.

In addition, the untargeted metabolomics analysis is discussed after the targeted structural elucidation of two “major classes of secondary metabolites”. I understand organizing the manuscript in this way. However, considering that the two compounds isolated were the ones tested for biological activity, which is discussed last, the untargeted analysis is rather repetitive and orthogonal to the main work in the paper. If the authors are set on including all of the data in a single manuscript, I suggest discussing the untargeted approach first, using the interaction to justify focusing on the two main compounds, then discussing the biological activity of the two compounds. Since the focus of the paper is actually the biological activity, it may not be necessary to discuss the full structural isolation and NMR elucidation of the two main compounds as they have been previously described.

>We are glad the reviewer understands why we organised it this way – to us it seems the most logical way to present the data. Further, certain comments above seem inconsistent with points raised below: specifically, above the reviewer states that as the focus is biology, and that less discussion is required for the structure determination as they are known compounds. Indeed, that is why we gave limited structural data for the known compounds. However, below the reviewer suggests there may be issues with the structure determination/identity of these compounds and asks for extensive data to be provided. This includes MS2 data, and data for MRM spectra within the context of our quantification experiments. These last two requests are excessive in our opinion but we have responded to the various requests as appropriate.

Further discussion of the BGCs and/or biological activity of the compounds would enhance the manuscript.

>We do not think the BGCs merit further discussion, the genomes were sequenced to determine the presence or absence of appropriate BGCs, and this was achieved using BlastX2.2.31+ with amino

acid sequences from the published, annotated BGCs as stated in the Methods. The biological activity of the compounds is already a major focus of the paper.

2. Data is not consistently quantified in the manuscript. For example, Figure 2 shows the absorbance profiles of the extracts of the interaction of *L. gongylophorus* and *E. weberi* plus their control axenic cultures. In the text, the authors state “Both 1 & 2 were absent in extracts of the healthy food fungus, but were elevated approx. three-fold when compared to axenic cultures of *E. weberi* (Fig. 2).” Based upon the figure presented, I cannot draw that conclusion. This data can and should be quantified by using area under the curve values. Further, in the methods section the statement “Quantitative analysis was performed...” indicates that these values were calculated, although not included.

> We believe it is evident from the HPLC traces shown that compounds 1 & 2 are absent from the extracts from the healthy food fungus (lower green trace). In addition, we have added text to state that this was confirmed by analysis of the LC(UV)MS spectra for this experiment using comparison to the isolated standards of 1 & 2. Please note that this point was already made in the Materials & Methods section, and clearly states that areas under peaks were measured. However, for clarity we have included in the AUC data in the M&M and SI Fig 1 legends. We also now include the individual fold-increase numbers for 1 and 2 in the main text and the legend for Supp Fig 1.

>Also note that these write ups are more extensive – see response to later comment below regarding the analytical methodology.

>Finally, we have swapped the co-culture experiment that was in the SI (Supp Fig 1) into the main text – now it is Fig 2. The old Fig 2 is now Supp Fig 1. On reprising our data, we decided that it was most sensible to include the most compelling of the two independently run co-culture experiments in the main text. This does not, in any way, change the conclusions of the manuscript.

Another example is Figure 5. The MICs of the two main compounds against two strains of *Pseudonocardia* were measured via resazurin. However, the figure displays a photograph of a single replicate of the MIC measurements. Visually, even via pdf on a monitor, it is hard to distinguish the pink/purple/blue colors of the photograph. Resazurin conversion to resorufin can be quantified using fluorescence (or absorbance) measurement.

> We updated the figure and hope that the colour differences can now be seen clearly. This includes the triplicate data in the figure.

3. It would be helpful if the authors included additional data in their supplemental files. Inclusion of NMR spectra (versus the listed ppm shifts)

>Images of relevant NMR spectra have been included in the Supplementary Info (Supplementary Figures 14-31). We did not include these originally at it is usually standard and sufficient to provide key basic data and state that this agrees with published literature (giving appropriate references).

and MS data (MS2 spectra, MRM quantitation, etc.) would provide stronger support of the structural conclusions and quantification measurements.

>Given the inclusion of NMR spectra data, on top of other characterization info, it seems unnecessary to provide images of additional MS data to support the identification of compounds 1-8.

Concerning the MRM quantitation, we provided an extremely detailed section in the M&M (entitled ‘MS based quantitation of 1 and 2’) which will allow anyone to repeat this experiment; this section also provides all details of the MS and other equipment as noted in response to a question below. Similarly, as is also noted in response to a question below, we included a very appropriate internal standard for these experiments. Moreover, full details of mass transitions used (including additional ones examined) plus the relevant cone voltage, collision energy and dwell times are also provided in Supplementary Table 8. Finally, as also noted we used isolated and validated standards of the target

molecules for which LCMS/MS data was of course available. This seems adequate and standard to us, so could the reviewer please explain what additional information they think is needed?

I request this because the settings for molecular networking are quite broad for high-resolution mass spectrometry data. The precursor ion mass tolerance of 2.0 Da and the fragment ion mass tolerance of 0.5 Da are the default settings for molecular networking. This means that the precursor ion mass window is 4.0 Da. Typically for HRMS data, assuming 10 ppm error, a precursor ion mass tolerance of 0.05 Da (precursor ion mass window of 0.1 Da) should be sufficient across the entire measured mass range. In addition, the cosine score is set to 0.5 and the minimum matched paired peaks is set to two. The cosine score is rather low (default setting is 0.7), but the concern is really the minimum matched paired peaks. This means that in order for two MS/MS spectra to show similarity, the MS/MS spectra only have to contain two overlapping or related peaks. These settings may have been required in order to ensure that the compounds of interest networked, but the MS/MS spectra should be included in the supplemental information as lower settings for cosine and minimum matched paired peaks are more likely to allow unrelated MS/MS spectra to cluster. Inclusion of the MS² spectra would allow validation of the networking results.

>We thank this reviewer for their detailed analysis and their sound understanding of the networking algorithm. We agree with the reviewer about a rather wide mass tolerance used. Since we checked all hits manually for consistency (retention time, high resolution MS signal etc., as for many of the compounds we had isolated standards as noted above), we do not feel that this wide filter puts the validity of our study at risk.

A minimum matched paired peaks of two is considered by the reviewer as a very low value and we can understand the concerns related. The reason we have chosen a value of two is based on the fact that we detected a number of minor Shearinine derivatives occurring at very low levels. Given that we performed the analysis on an ion trap TOF system, and not on a much more sensitive instrument such as an Orbitrap analyzer, only few MS² Signals were detectable for the lowest analyte levels. If we had set the minimum matched peaks to a higher level, many hits would not have been included in our results, leading to false negative results. In order to make up for such a wide filter, again, we manually checked our hits for consistency, based on the retention time and the high resolution MS signal of their parent ions.

On this basis we do not feel individual MS/MS spectra are required. However, we have included additional cosine 'scores' in the ETP and terpene-indole alkaloid subnetwork figures as noted below.

4. Figure 4 is illegible. It may make more sense to move the large heatmap and molecular network to the supplemental information. Supplemental Table 4 would provide a better visualization of the heatmap and the sub-network of the melinacidin/chetracin molecular family shown in Supplemental Figure 5 would be more useful in the main text. It would be helpful if the authors labeled the cosine scores between two connected nodes for the enlarged figures of the molecular families. It may be beneficial if the sub-network for the terpenoid indole alkaloids was also included.

>We agree with reviewer 2 and have switched Fig 4 for Supp Table 4 in the main text, and edited this table and legend slightly for improved clarity; it is now called Table 1.

In conjunction with this we have simplified and improved what was Figure 4 and located this in the Supplementary information as Supp Figure 5 as it provides a global metabolic view of these strain. Furthermore, we have added the requested cosine scores to what was Supp Figure 5, which shows the ETP sub-network, and improved the annotation of this figure; as suggested by the reviewer we have included this in the main text where it is now called Figure 4. Similarly, we have produced a figure for the terpene-indole alkaloid sub-network with cosine scores and relevant annotations as Supp Figure 6.

Minor concerns:

General:

The authors use standard error for their statistical calculations, which suggests that they aim to compare the averages between the different experimental groups. However, very few of these comparisons are made in the text or denoted in the bar graphs.

>Significance tests between groups have been carried out using Dunn's non-parametric multiple comparisons tests. Adjusted P-values have now been quoted in the corresponding text and the graph updated to show significant comparisons.

ETPs is used throughout, but it is not defined.

>We define ETPs when it is first mentioned in the Results, sub-section 'Identification of the *Escovopsis* virulence factors.'

ppm error should be reported for (putatively) annotated compounds

>These data have been included in the Supplementary Information (Notes) alongside the reported calculated and observed HR-EIS-MS data for compounds 1 to 8.

There is a lot of gene information included in the manuscript, but unfortunately, is not discussed at length. Gene numbers should be included in genome/BGC annotations, if available.

> The genome sequencing in our study was performed solely to identify these BGCs (see detailed response to reviewer 3 below). We did not annotate the genomes so gene numbers are not available. However, we have added gene numbers from the reference *E weberi* genome (de Man et al 2017).

Supplementary Figure 2 is missing some of the NOESY data (arrows without shifts).

> These data have been added to the figure

Supplementary Figure 11 appears to have had conversion issues.

> We have supplied a new version of Supp Fig 11.

It is unclear why yohimbine was used as an internal standard for quantification. Was this because it is also an indole alkaloid?

>Yes, and because it is commercially available. Moreover, there are numerous publications regarding analytical MS based studies for the identification and quantification of this molecule, meaning its behaviour is well understood. It has also been employed as an internal control in several published studies. These reasons make it an appropriate internal reference.

Introduction:

Paragraph 1

Lines 6-7: "...immunity of their colonies. Well known examples are the..."

Unclear. I am assuming the examples are eluding to fungi that specialize on ants as host.

>Yes, you are correct

Paragraph 2

Lines 7-9: “The gongylidia serve as the sole...fresh leaf pulp.”

This sentence does not make sense. May want to consider rewording as “The gongylidia have two functions: the sole food source for ant larvae and as a source of enzymes for leaf pulp decomposition.” Or something similar.

> OK, we have changed this to: ‘The gongylidia serve two functions: they are the sole food source for the ant larvae, but are also ingested by the ant workers to transmit fungal decomposition enzymes to the ant fecal fluid which is deposited on the fresh leaf pulp’

Paragraph 3: This paragraph has abrupt changes between commensal and invasive microbes. May want to consider starting a new paragraph at “It has been estimated...”

> OK, we have done this.

Paragraph 3

Lines 12-14: “In *A. echinator*, such events reflect a failure...” This sentence should be simplified.

> OK, we have simplified this sentence to: ‘In *A. echinator*, such events reflect a failure of the *Pseudonocardia* biofilm defences. They occur frequently when colonies are freshly excavated and kept in captivity because the ants cannot dump their waste away from the garden.’

Lines 14-15: “The colony collapse that typically follows...” Follows what event?

> Infection by *Escovopsis*. We have changed this sentence to: ‘The resulting colony collapse suggests that *Escovopsis* hyphae produce metabolites that affect worker ant behavior and which, at high enough concentrations, are lethal to the ants.’

Paragraph 4

Line 1: secondary or specialized is sufficient.

> OK, we removed ‘specialized’

Line 3-5: simplify by removing both usages of “both”

>OK

Line 10: *E. weberi* should be spelled out for first use.

> It is spelled out in the Line 1 of that paragraph: ‘Here we identify two secondary metabolites that are upregulated during *Escovopsis weberi*’

Results:

Paragraph 1

Line 3: “...and comparing to axenic cultures...”

Comparing what features of the co-culture experiment to the axenic cultures?

> We have changed this sentence to read: ‘comparing the secondary metabolome to that of axenic cultures of the two organisms.’

Lines 5-6: “...of two major and a range of minor metabolites...”

What defines major and minor? Amount produced? Bioactivity?

> Amount produced. This is clearly articulated in the preceding part of this sentence: ‘Chemical profiling of extracts taken from the resulting plates, using UPLC coupled with high-resolution mass

spectrometry (HRMS), revealed the presence of two major and a range of minor metabolites produced during pathogenesis’.

Line 6-7: “...had signals of $m/z = 729.0937...$ ”

Change “of” to “at” and remove “=”

> OK, we have changed this.

Line 9: “Essentially identical...”

What does this mean? How can this conclusion be drawn when the gradients were clearly different and the fold change is 7 versus 3.

> We have changed this sentence to: ‘This experiment was carried out independently in the Hutchings and Wilkinson lab (Norwich), but the results are similar to those from the experiment carried out in Jena (Fig. 2, main text). (Supplementary Fig. 1).’ While the experimental set ups were different, and fold-change for **1** and **2** were also different between experiments, they do show the same trend and provide independent support for each other and the conclusions drawn.

Please note that the Fig2 and Supp Fig 1 have been swapped as discussed above.

Moreover, as the work was done independently (we started collaborating after the initial experiments were done in each location), each group had a different analytical system. However, as the identity and quantification of the compounds responsible for the HPLC peaks were subsequently linked to the compounds by isolation and structure elucidation independently, the fact that analytical systems vary is irrelevant.

Paragraph 2

Lines 1-3: This sentence does not make sense. Presumably, a large scale cultivation of the axenic *E. weberi* was performed to isolate the compounds.

> Agreed, we have changed this to: ‘For structural elucidation we isolated **1** and **2** from large-scale axenic cultures of *E. weberi* strain A grown on PGA plates.’

Line 7: Replace “nature” with “compound”.

> OK we have changed this.

Paragraph 3:

Line 15: Reference is needed after “...same Panamanian field site.”

> OK, citation has been added.

Paragraph 4:

Line 4: Remove “that we”

> OK we have changed this.

Line 7: Change to “UPLC-MS/MS” (may need to define MS/MS)

> OK we have changed this.

Line 19: Remove “finally”

> OK we have removed this.

Paragraph 5

Lines 1-2: Remove “that we aimed to unambiguously identify.”

> OK we have removed this.

Line 2: “Production levels were sufficiently...”

Production levels of what compounds?

> This sentence goes on to name the compounds.

Paragraph 6

Line 5: “The metabolomics data clearly showed...”

The heatmap associated with this conclusion is not clear. See major concern #4.

> This is addressed above in response to the last ‘Major concern #4’.

Paragraph 7

Lines 11-12: Remove “fungal antiSMASH”; Add “from fungus” after BGCs.

> OK we have removed this.

Line 15: Removed “could”; change tense of “identify” to “identified”

> OK we have made these changes.

Paragraph 9

Line 10-13: “...famous example is produced...”

This sentence is incomplete. What is the example fungal virulence factor?

> The identity of the compound is not known. We have modified the following sentence to make this clear.

Paragraph 10

Line 3: Remove which could be.

> OK

Paragraph 11

Line 8-9: Reference is needed after ...rather than poisoning.”

> OK

Line 9: Remove “then”

> OK we have removed this.

Discussion

Paragraph 1

Line 6: Remove “widespread”

> OK we have removed this.

Materials and Methods

Paragraph 1: Were appropriate permits gathered (if necessary)?

> Yes, see acknowledgements.

Paragraph 2

Line 3: To what does “them” refer?

> We agree this was unclear and have changed this sentence to: ‘For strains originating from *A. echinator* colonies, *Escovopsis* was isolated from the waste dumps of lab colonies, or by incubating sections of fungus garden, in Petri dishes with moist cotton wool.’

Line 3-4: Change “amplify” to “amplification”; What gene was used?

> We have changed this to: ‘Strain identity was verified by PCR amplification and sequencing of the 18S gene using the primer set 18S1A and 18S564 (Supplementary Table 7).’

Line 8-10: This sentence does not make sense and would not be reproducible.

> We have changed this to: ‘Stocks were made from lawns of each purified *Escovopsis* strain grown on 25 ml PGA agar plates. The spores were harvested by applying 2 ml of sterile glycerol solution (20%, v/v) to the surface of the plate and agitating using sterile cotton buds. The spore suspension from each plate was then transferred to a 2 ml screw cap tubes and stored at -80 °C.’

Paragraph 4: Considering that two experiments measuring the two major metabolites were performed in different labs under clearly different gradients (figure 2 and suppl. figure 1), there should be two sets of HPLC methods.

> We have edited the MM and Supp Fig 1 to clearly give the conditions for the two different co-cultivation experiments, including the relevant analytical HPLC, and quantification; this is noted above in responses to earlier questions; again, please note that Fig 2 and Supp Fig 1 have been swapped for impact and clarity, but this does not in any way change the conclusions of the manuscript. We thank reviewer 2 for drawing this to our attention, we had missed this mix up of information from the two independent labs.

Paragraph 5: For the HRMS, was internal and/or external calibration used? What was the eV setting used for CID?

> For HRMS an external calibrant was used – the text has been amended by addition of the following text: ‘The instrument was calibrated using sodium trifluoroacetate cluster ions according to the manufacturer’s instructions.’

In contrast to other vendors Shimadzu does not give direct access to the magnitude of collision energies used (which in fact would be of limited use, due to differences in the design of the collision cell among different vendors). Instead, a certain percentage can be adjusted by the user. We have already provided this information at the end of the section ‘Spectroscopy’: The detector voltage of the TOF mass analyzer was set to 1.66 kV and the collision-induced dissociation (CID) energy to 50%.

Paragraph 7

Lines 5-6: “Samples were analyzed the standard method described in Spectroscopy section above.” What does this mean?

> Changed to ‘The supernatant was filtered, evaporated and the residue dissolved in methanol. Samples were analyzed as described in the Spectroscopy section, above.’

Paragraph 8

Line 2: Change “transferred” to “converted”

> OK, we’ve changed this.

Paragraph 10: It is not clear how the movement of each ant was quantified. Was this analysis performed manually or through a program?

> We have amended the methods to make it clear this was done by eye and time spent stationary was timed and measured in seconds.

Line 2: Remove “then”

> OK, we’ve removed this.

Line 15: Remove “at the latest”

> OK, we’ve removed this.

Paragraph 12: For the MRM method, what mass spectrometer was used as the UPLC system is Waters not Shimadzu.

> This information is already present at the start of the method development paragraph; the reviewer must have missed it. For clarity, it was a Xevo TQS tandem quadrupole mass spectrometer.

Line 4: Remove “additionally”

> OK, we’ve removed this.

Reviewer #3 (Remarks to the Author):

The manuscript provides new insight into the chemical warfare of leafcutter ant symbionts and pathogenic *Escovopsis*. In the manuscript, the authors present direct genomic and metabolomic evidence describing the virulence factors produced by invading fungus and evaluate their activity on the mutualistic *Pseudonocardia* and host ant. The authors complete a comprehensive study identifying, and quantifying these compounds and their congeners, and point to a genomic origin for these compounds. However, their genomic analysis is less comprehensive with insufficient evidence presented to objectively evaluate their arguments. Moreover, this analysis is peripheral to the main conclusions of the work with no justification or deeper analysis. Overall, the manuscript is well written with few errors and will provide valuable information to entomologists working with leafcutter ants.

> We agree that whole genome analysis is peripheral to this study. The genomes were sequenced simply to look for the BGCs for shearinine, emodin and melinacidin (ETP) like metabolites, and we state this clearly in the opening line of that results section: “To gain deeper insight into the secondary metabolism of the fungal pathogen, we sequenced the genomes of the six *Escovopsis* strains A-F used in this study.” On this basis we perform sufficient analysis to verify that the various compound class BGCs are present. More detailed analysis of the biosynthetic potential of these strains, from both genomics and chemical approaches, will be the subject of future work in our labs.

Specific questions

It is unclear how strong the BCG evidence is as the level of homology is not discussed. How similar

are these clusters to each other and the known/characterized cluster? How do you assess the accuracy of your characterization e.g. EmoH == decarboxylases?

> Genes in each BGC were identified using BlastX with query sequences from the published, annotated BGCs. We have made this clear in the methods section.

Do the differences in clusters explain the differences in congener profiles? Ie. More divergent gene clusters preferentially produce another congener?

> This is difficult to predict based purely on genomics data, and would be very speculative.

It is unclear what Figure 4A is depicting both in image resolution and data representation. What metabolites do the rows represent? What do the columns represent? At first glance, these look like they represent different strains of *E. weberi* but these strains have each been subdivided into distinct columns. How was the data organized? why are the shearinines not together, for example?

> We have removed this figure, as suggested by Reviewer 2, and replaced it with Supp Table 4. Hopefully this is now clear.

Supp Fig 7 – Strains E, F, G mislabelled

> Thanks, we have corrected this.

Why was *tef1* alone used for phylogenetic analysis of *Escovopsis* as opposed to the more common ITS1/ITS2 standard for fungal species? See Schoch, C. L. et al. Nuclear ribosomal internal transcribed spacer (ITS) region as a universal DNA barcode marker for Fungi. PNAS 109, 6241–6246 (2012); Meirelles LA, Montoya QV, Solomon SE, Rodrigues A (2015) New Light on the Systematics of Fungi Associated with Attine Ant Gardens and the Description of *Escovopsis kreiselii* sp. nov. PLoS ONE 10(1): e0112067, ref 49

> We used *tef1* because it was used in the *Escovopsis* phylogenetics paper published by Meirelles et al 2015. That study did also use the ITS sequence but this is missing from the reference *E. weberi* genome (we call strain G) published by de Man et al 2017 and so we could not use ITS to compare all the genome sequenced *Escovopsis* strains. However, we have now performed the analysis on the strains used in our study (A-F) using both the *tef1* and ITS sequences and updated the phylogenetic analyses to include both (Supplementary Table 5 and Supplementary Figure 8). The conclusion remains the same.

"with five species recognized" change to "with five classified species"

> OK, we have changed this

How does your clade analysis align with that in ref 49 as your phylogenetic analysis use different reference genes? Also, ref 49 defines no *E. weberi* species as a member of the 9 proposed clades (para. On Genome sequencing)

> See above. We have now performed the analysis on strains A-F using both the ITS and *tef1* sequences and updated the phylogenetic tree in Supplementary Table 5 and Supplementary Figure 8.

There are broken characters in Sup Fig 11 (boxes).

> Thanks, we have replaced this figure

How do you determine that the species you have isolated are actually *E. weberi* vs and *Escovopsis* with 'weberi morphology' as described in ref 49?

> Through phylogenetic analysis using the *tefl* gene and ITS DNA sequences and comparison to Merilles et al 2015, as described.

REVIEWERS' COMMENTS:

Reviewer #2 (Remarks to the Author):

I think the author revisions addressing my concerns regarding the manuscript are appropriate. I appreciate the authors taking the time to streamline the narrative, improve the figures, and add additional detail to figures, methods, and supplemental information.

Reviewer #3 (Remarks to the Author):

The authors have adequately responded to most reviewer comments but one shared by two reviewers about the BCG analysis. While the authors are quite clear in their methods for the analysis, they provide no actual data or evidence for the reviewers to evaluate their claims. Namely, they annotate genomes and make claims to the presence of BCGs and gene function without providing a single E value, % query coverage, % identity or any other metric/cutoff that would allow an educated reviewer to evaluate the veracity of these claims.

This reviewer agrees that the genomic analysis is tangential to the main thrust of the paper but if an author makes a claim on the basis of genomic evidence, the onus is on the author to provide some supporting evidence for said claim.

NCOMMS-17-29297A revisions

Reviewer #2 (Remarks to the Author):

I think the author revisions addressing my concerns regarding the manuscript are appropriate. I appreciate the authors taking the time to streamline the narrative, improve the figures, and add additional detail to figures, methods, and supplemental information.

We are pleased that reviewer 2 considered our revisions to be satisfactory and appreciate that their efforts have helped to improve our manuscript.

Reviewer #3 (Remarks to the Author):

The authors have adequately responded to most reviewer comments but one shared by two reviewers about the BCG analysis. While the authors are quite clear in their methods for the analysis, they provide no actual data or evidence for the reviewers to evaluate their claims. Namely, they annotate genomes and make claims to the presence of BCGs and gene function without providing a single E value, % query coverage, % identity or any other metric/cutoff that would allow an educated reviewer to evaluate the veracity of these claims.

This reviewer agrees that the genomic analysis is tangential to the main thrust of the paper but if an author makes a claim on the basis of genomic evidence, the onus is on the author to provide some supporting evidence for said claim.

We agree with reviewer 3 that the onus is on us to provide supporting evidence for our claims regarding the biosynthetic gene clusters (BGCs) reported in our paper.

We first include an extra table in the supplementary information (now SI Table 3) which provides a detailed analysis (BlastP) of the terpene-indole alkaloid BGC reported for the previously published strain G; this include predicted function, E values and % identity data as suggested by reviewer 3 and by the Nat Comms editorial team in their previous email – these are then referred to in the main text, again as suggested by the editorial team. These data are readily generated as there are both genome and transcriptome data available for this strain – we provide this as a reference analysis.

We then provide a second new table (SI Table 4) which provides a BlastX analysis of the protein sequences from reference strain G (see above, SI Table 3) and the nucleotide sequences for the BGCs from strains A-F. Once again, these data include metrics as suggested by the reviewer and editors.

These analyses are then repeated for the ETP encoding BGCs (now SI Tables 6 and 7), and for the emodin encoding BGCs (now SI Tables 9 and 10).